# A General Theory of Correct, Incorrect, and Extrinsic Equivariance

**Dian Wang**[1]   **Xupeng Zhu**[1]   **Jung Yeon Park**[1]   **Mingxi Jia**[2*]   **Guanang Su**[3*]
**Robert Platt**[1]   **Robin Walters**[1]
[1]Northeastern University   [2]Brown University   [3]University of Minnesota
{wang.dian,zhu.xup,park.jungy,r.platt,r.walters}@northeastern.edu
mingxi_jia@brown.edu   su000265@umn.edu

## Abstract

Although equivariant machine learning has proven effective at many tasks, success depends heavily on the assumption that the ground truth function is symmetric over the entire domain matching the symmetry in an equivariant neural network. A missing piece in the equivariant learning literature is the analysis of equivariant networks when symmetry exists only partially in the domain. In this work, we present a general theory for such a situation. We propose pointwise definitions of correct, incorrect, and extrinsic equivariance, which allow us to quantify continuously the degree of each type of equivariance a function displays. We then study the impact of various degrees of incorrect or extrinsic symmetry on model error. We prove error lower bounds for invariant or equivariant networks in classification or regression settings with partially incorrect symmetry. We also analyze the potentially harmful effects of extrinsic equivariance. Experiments validate these results in three different environments.

## 1   Introduction

Equivariant neural networks [9, 10] have proven to be an effective way to improve generalization and sample efficiency in many machine learning tasks. This is accomplished by encoding task-level symmetry into the structure of the network architecture so that the model does not need to explicitly learn the symmetry from the data. However, encoding a fixed type of symmetry like this can be limiting when the model symmetry does not exactly match the symmetry of the underlying function being modeled, i.e., when there is a symmetry mismatch. For example, consider the digit image classification task. Is it helpful to model this problem using

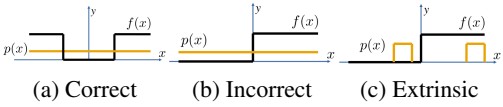

(a) Correct          (b) Incorrect          (c) Extrinsic

Figure 1: An example of correct, incorrect, and extrinsic equivariance. The ground truth function $f(x)$ is shown in black and its probability density function $p(x)$ is shown in orange. If we model $f(x)$ using a $G$-invariant network where $G$ is a reflection group that negates $x$, different $f(x)$ and $p(x)$ will lead to correct, incorrect, and extrinsic equivariance. See Section 3 for details.

a model that is invariant to 180-degree rotation of the image? For some digits, the label is invariant (e.g., **0** and **8**). However, for other digits, the label changes under rotation (e.g., **6** and **9**), suggesting that a rotationally symmetric model would be inappropriate here. However, recent work [58] suggests that this is not necessarily the case – symmetric models are sometimes helpful even when a symmetry mismatch exists between the problem and model. This raises the question – do the advantages obtained by using a symmetric model outweigh the errors introduced by the symmetry mismatch?

---

*Work done as students at Northeastern University

37th Conference on Neural Information Processing Systems (NeurIPS 2023).

This paper makes four main contributions towards this problem. First, this paper extends the definitions for types of model symmetry with respect to true symmetry introduced in Wang et al. [58]. They classify models as having *correct*, *incorrect*, or *extrinsic equivariance* (see Figure 1), where correct means the ground truth function has the same symmetry as the equivariant model, incorrect means the ground truth function disagrees with the symmetry in the model, and extrinsic means the model symmetry transforms in-distribution data to out-of-distribution data. We generalize this system into a continuum of equivariance types to reflect the fact that a single task may have different proportions of correct, incorrect, and extrinsic symmetry across its domain. For example, in the digit classification task, **0** has correct equivariance, **6** has incorrect equivariance, and **4** has extrinsic equivariance.

Our second contribution is to introduce an analytical lower bound on model error in classification tasks resulting from incorrect model symmetry. This result can help guide model selection by quantifying error resulting from incorrect equivariance constraints. Our result generalizes that of Wang et al. [58] by removing the simplifying assumption that data density over the domain is group invariant. We prove the minimum error of an invariant classifier can be realized by assigning all data points in the same group orbit the label with the majority of the data density (Theorem 4.3).

Our third contribution is to develop new lower bounds on the $L_2$ error for regression tasks in terms of the variance of the function to be modeled over the orbit of the symmetry group. Like our classification bound, this bound can assist in model selection in situations with symmetry mismatch.

Fourth, in contrast to Wang et al. [58] who show benefits of extrinsic equivariance, we theoretically demonstrate its potential harm. We perform experiments documenting the error rate across the correct-extrinsic continuum. Finally, we perform empirical studies illustrating the ideas of the paper and showing that the lower bounds obtained in our analysis appear tight in practice. This suggests our analysis can assist practitioners select symmetry groups appropriate for a given problem setting. Our code is available at `https://github.com/pointW/ext_theory`.

## 2 Related Work

**Equivariant Learning.** Originally used for exploiting symmetry in image domains [9, 10], equivariant learning has been very successful in various tasks including molecular dynamics [2, 4], particle physics [6], fluid dynamics [59], trajectory prediction [53], pose estimation [31, 26, 32], shape completion [7], robotics [48, 65, 21, 55, 49, 41, 23, 22, 45] and reinforcement learning [52, 54, 56, 39, 63]. However, most prior work assumes that the symmetry of the ground truth function is perfectly known and matches the model symmetry. Wang et al. [58] go further and define correct, incorrect, and extrinsic equivariance to classify the relationship between model symmetry and domain symmetry. However, they do not discuss the possible combinations of the three categories, and limit their theory to a compact group and invariant classification. Our work extends [58] and allows for a continuum of equivariance types and analyzes error bounds in a more general setup.

**Symmetric Representation Learning.** Various works have proposed learning symmetric representations, using transforming autoencoders [20], restricted Boltzmann machines [50], and equivariant descriptors [47]. In particular, [30] shows that convolutional neural networks implicitly learn representations that are equivariant to rotations, flips, and translations, suggesting that symmetric representations are important inductive biases. Other works have considered learning symmetry-aware features using disentanglement [43], projection mapping [25], equivariance constraints [35], separation into invariant and equivariant parts [61] or subgroups [34]. Park et al. [42] propose learning a symmetric encoder that maps to equivariant features and Dangovski et al. [13] learn features that are sensitive and insensitive to different group representations. Other works assume no prior knowledge of symmetry and learn it from data [3, 64, 14, 40]. In particular, Moskalev et al. [40] estimate the difference between the true latent symmetry and the learned symmetry. Similarly, our work considers a gap between the true symmetry and model symmetry and theoretically analyze its effects on error.

**Theory of Equivariant Learning.** There are several lines of work on the theory of equivariant learning. Kondor and Trivedi [27] prove that convolutions are sufficient and necessary for equivariance of scalar fields on compact groups, later generalized to the steerable case by Cohen et al. [11]. Certain equivariant networks have been proved to be universal in that such networks can approximate any $G$-equivariant function [36, 62]. Another line of work has considered equivariant networks in terms of generalization error. Abu-Mostafa [1] show that an invariant model has a VC dimension less than

or equal to that of a non-equivariant model. Other works studied the generalization error of invariant classifiers by decomposing the input space [51, 46]. Elesedy and Zaidi [17] quantify a generalization benefit for equivariant linear models using the notion of symmetric and anti-symmetric spaces. A PAC Bayes approach was used for generalization bounds of equivariant models [5, 33]. Our work is complimentary to these and quantifies approximation error for equivariant model classes.

**Data Augmentation.** Some methods use data augmentation [29, 28] to encourage the network to learn invariance with respect to transformations defined by the augmentation function [8]. Recent works have explored class-specific [19, 44] and instance-specific [38] data augmentation methods to further boost training by avoiding the potential error caused by a uniform augmentation function. Those methods can be viewed as applying data augmentation where pointwise correct or extrinsic invariance exist, while avoiding incorrect invariance.

## 3 Preliminaries

**Problem Statement.** Consider a function $f: X \to Y$. Let $p: X \to \mathbb{R}$ be the probability density function of the domain $X$. We assume that there is no distribution shift during testing, i.e., $p$ is always the underlying distribution during training and testing. The goal for a model class $\{h: X \to Y\}$ is to fit the function $f$ by minimizing an error function $\mathrm{err}(h)$. We assume the model class $\{h\}$ is arbitrarily expressive except that it is constrained to be equivariant with respect to a group $G$. Let $\mathbb{1}$ be an indicator function that equals to 1 if the condition is satisfied and 0 otherwise. In classification, $\mathrm{err}(h)$ is the classification error rate; for regression tasks, the error function is a $L_2$ norm function,

$$\mathrm{err}_{\mathrm{cls}}(h) = \mathbb{E}_{x \sim p}[\mathbb{1}(f(x) \neq h(x))], \qquad \mathrm{err}_{\mathrm{reg}}(h) = \mathbb{E}_{x \sim p}[||h(x) - f(x)||_2^2]. \qquad (1)$$

**Equivariant Function.** A function $f : X \to Y$ is equivariant with respect to a symmetry group $G$ if it commutes with the group transformation $g \in G$, $f(gx) = gf(x)$, where $g$ acts on $x \in X$ through the representation $\rho_X(g)$; $g$ acts on $y \in Y$ through the representation $\rho_Y(g)$.

### 3.1 Correct, Incorrect, and Extrinsic Equivariance.

Consider a model $h$ which is equivariant with respect to a group $G$. Since real-world data rarely exactly conforms to model assumptions, in practice there may often be a gap between the symmetry of the model and the ground truth function. Wang et al. [58] propose a three-way classification which describes the relationship between the symmetry of $f$ and the symmetry of $h$. In this system, $h$ has correct equivariance, incorrect equivariance, or extrinsic equivariance with respect to $f$.

**Definition 3.1** (Correct Equivariance). For all $x \in X, g \in G$ where $p(x) > 0$, if $p(gx) > 0$ and $f(gx) = gf(x)$, $h$ has *correct equivariance* with respect to $f$.

**Definition 3.2** (Incorrect Equivariance). If there exist $x \in X, g \in G$ such that $p(x) > 0, p(gx > 0)$, but $f(gx) \neq gf(x)$, $h$ has *incorrect equivariance* with respect to $f$.

**Definition 3.3** (Extrinsic Equivariance). For all $x \in X, g \in G$ where $p(x) > 0$, if $p(gx) = 0$, $h$ has *extrinsic equivariance* with respect to $f$.

**Example 3.4.** Consider a binary classification task where $X = \mathbb{R}$ and $Y = \{0, 1\}$. If the model $h$ is invariant to a reflection group $G$ where the group element $g \in G$ acts on $x \in X$ by $gx = -x$, Figure 1 shows examples when correct, incorrect, or extrinsic equivariance is satisfied.

### 3.2 Pointwise Equivariance Type.

Although Definitions 3.1- 3.3 are self-contained, they do not consider the mixture of different equivariance types in a single function. In other words, an equivariant model can have correct, incorrect, and extrinsic equivariance in different subsets of the domain. To overcome this issue, we define pointwise correct, incorrect, and extrinsic equivariance, which is a generalization of the prior work.

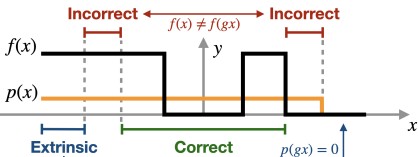

**Definition 3.5** (Pointwise Correct Equivariance). For $g \in G$ and $x \in X$ where $p(x) \neq 0$, if $p(gx) \neq 0$ and $f(gx) = gf(x)$, $h$ has correct equivariance with respect to $f$ at $x$ under transformation $g$.

Figure 2: Example of pointwise correct, incorrect, and extrinsic equivariance in a binary classification task. $f(x)$ is in black and $p(x)$ is in orange. $G$ is a reflection group that negates $x$.

**Definition 3.6** (Pointwise Incorrect Equivariance). For $g \in G$ and $x \in X$ where $p(x) \neq 0$, if $p(gx) \neq 0$ and $f(gx) \neq gf(x)$, $h$ has incorrect equivariance with respect to $f$ at $x$ under transformation $g$.

**Definition 3.7** (Pointwise Extrinsic Equivariance). For $g \in G$ and $x \in X$ where $p(x) \neq 0$, if $p(gx) = 0$, $h$ has extrinsic equivariance with respect to $f$ at $x$ under transformation $g$.

Notice that the definitions of pointwise correct, incorrect, and extrinsic equivariance are mutually exclusive, i.e., a pair $(x, g)$ can only have one of the three properties. The pointwise definitions are generalizations of the global Definitions 3.1- 3.3. For example, when pointwise correct equivariance holds for all $x \in X$ and $g \in G$, Definition 3.1 is satisfied.

**Example 3.8** (Example of Pointwise Correct, Incorrect, and Extrinsic Equivariance). Consider the same binary classification task in Example 3.4. Figure 2 shows $f(x)$, $g(x)$, and four subsets of $X$ where pointwise correct, incorrect, or extrinsic holds. For $x$ in the correct section (green), $p(x) > 0, p(gx) > 0, f(x) = f(gx)$. For $x$ in the incorrect sections (red), $p(x) > 0, p(gx) > 0, f(x) \neq f(gx)$. For $x$ in the extrinsic section (blue), $p(x) > 0, p(gx) = 0$.

**Definition 3.9** (Correct, Incorrect, and Extrinsic Sets). The Correct Set $C \subseteq X \times G$ is a subset of $X \times G$ where pointwise correct equivariance holds for all $(x, g) \in C$. Similarly, the Incorrect Set $I$ and the Extrinsic Set $E$ are subsets where incorrect equivariance or extrinsic equivariance holds for all elements in the subset. Denote $U \subseteq X \times G$ as the Undefined Set where $\forall (x, g) \in U, p(x) = 0$. By definition we have $X \times G = C \amalg I \amalg E \amalg U$, where $\amalg$ denotes a disjoint union.

# 4   Approximation Error Lower Bound from Incorrect Equivariance

Studying the theoretical error lower bound of an equivariant network is essential for model selection, especially when incorrect equivariance exists. Wang et al. [58] prove an error lower bound for an incorrect equivariant network, but their setting is limited to a classification task in the global situation of Definition 3.2 with a discrete group and an invariant density function. In this section, we find the lower bound of $\mathrm{err}(h)$ for an equivariant model $h$ in a general setting. To calculate such a lower bound, we first define the *fundamental domain* $F$ of $X$. Let $d$ be the dimension of a generic orbit of $G$ in $X$ and $n$ the dimension of $X$. Let $\nu$ be the $(n - d)$ dimensional Hausdorff measure in $X$.

**Definition 4.1** (Fundamental Domain). A closed subset $F$ of $X$ is called a fundamental domain of $G$ in $X$ if $X$ is the union of conjugates[2] of $F$, i.e., $X = \bigcup_{g \in G} gF$, and the intersection of any two conjugates has 0 measure under $\nu$.

We assume further that the set of all $x$ which lie in any pairwise intersection $\bigcup_{g_1 F \neq g_2 F} (g_1 F \cap g_2 F)$ has measure 0 under $\nu$. Let $Gx = \{gx : g \in G\}$ be the orbit of $x$, then $X$ can be written as the union of the orbits of all points in the fundamental domain $F$ as such $X = \bigcup_{x \in F} Gx$.

## 4.1   Lower Bound for Classification

We first show the lower bound of the error $\mathrm{err}_{\mathrm{cls}}(h)$ (Equation 1) given the invariant constraint in $h$: $h(gx) = h(x), g \in G$. In this section, the codomain $Y$ of $f$ is a finite set of possible labels. Since $h$ is $G$-invariant, $h$ has the same output for all inputs in an orbit $Gx$. We call the label that causes the minimal error inside the orbit the *majority label*[3], and define the error in the orbit as the *total dissent*.

**Definition 4.2** (Total Dissent). For the orbit $Gx$ of $x \in X$, the total dissent $k(Gx)$ is the integrated probability density of the elements in the orbit $Gx$ having a different label than the majority label

$$k(Gx) = \min_{y \in Y} \int_{Gx} p(z) \mathbb{1}(f(z) \neq y) dz. \tag{2}$$

We can also lift the integral to $G$ itself by introducing a factor $\alpha(x, g)$ to account for the Jacobian of the action map and size of the stabilizer of $x$. (See Appendix A.)

$$k(Gx) = \min_{y \in Y} \int_G p(gx) \mathbb{1}(f(gx) \neq y) \alpha(x, g) dg. \tag{3}$$

---

[2] A conjugate $gF$ is defined as $gF = \{gx | x \in F\}$.

[3] The majority label has more associated data than all other labels, but does not need to be more than 50%.

**Theorem 4.3.** $\mathrm{err}(h)$ *is lower bounded by* $\int_F k(Gx)dx$.

*Proof.* Rewriting the error function of Equation 1, we have

$$\mathrm{err}(h) = \int_X p(x)\mathbb{1}(f(x) \neq h(x))dx = \int_{x \in F} \int_{z \in Gx} p(z)\mathbb{1}(f(z) \neq h(z))dzdx, \quad (4)$$

using iterated integration (Appendix B) and Definition 4.1. We assume the measure of $F \cap gF$ is 0. Since $h(z)$ can only have a single label in orbit $Gx$, we can lower bound the inside integral as

$$\int_{z \in Gx} p(z)\mathbb{1}(f(z) \neq h(z))dz \geq \min_{y \in Y} \int_{z \in Gx} p(z)\mathbb{1}(f(z) \neq y)dz = k(Gx).$$

We obtain the claim by integrating over $F$. Notice that this is a tight lower bound assuming universal approximation. That is, there exists $h$ which realizes this lower bound. $\square$

We can express the total dissent in terms of the Incorrect Set $I$ (Definition 3.9).

**Proposition 4.4.** $k(Gx) = \min_{x' \in (Gx)^+} \int_G p(gx')\mathbb{1}((x', g) \in I)\alpha(x', g)dg$, *where* $(Gx)^+ = \{x_0 \in Gx | p(x_0) > 0\}$.

*Proof.* Consider Equation 3, since the minimum over $y$ is obtained for $y = f(x')$ for some $x' \in Gx$ such that $p(x') > 0$ (i.e., $x' \in (Gx)^+$),

$$k(Gx) = \min_{x' \in (Gx)^+} \int_G p(gx)\mathbb{1}(f(gx) \neq f(x'))\alpha(x, g)dg.$$

Since $x' \in Gx$, then $Gx' = Gx$ and we have $k(Gx) = k(Gx')$. Thus,

$$k(Gx) = \min_{x' \in (Gx)^+} \int_G p(gx')\mathbb{1}(f(gx') \neq f(x'))\alpha(x', g)dg$$

$$= \min_{x' \in (Gx)^+} \int_G p(gx')\mathbb{1}((x', g) \in I)\alpha(x', g)dg.$$

$\square$

**Example 4.5** (Lower bound example for a binary classification task using Proposition 4.4)**.**

Let $f \colon X \to \{0, 1\}$ be a binary classification function on $X = \{x_0, x_1, x_2, x_3\}$. Let $G = C_2 = \{e, r\}$ be the cyclic group of order two that permutes the elements in $X$. Figure 3 shows $X$, the label for each $x \in X$, and how $e, r \in G$ acts on $x \in X$. $\{x_0, x_3\}$ forms a fundamental domain $F$, and there are two orbits: $Gx_0 = \{x_0, x_1\}$ and $Gx_2 = \{x_2, x_3\}$. Since both $X$ and $G$ are discrete and $g \in G$ acts on $X$ through permutation, The lower bound can be written as $\mathrm{err}(h) \geq \sum_{x \in F} \min_{x' \in (Gx)^+} \sum_{g \in G} p(gx')\mathbb{1}((x', g) \in I)$. We can then calculate $\sum_{g \in G} p(gx')\mathbb{1}((x', g) \in I)$ for $x' \in X$: $x_0 : 0.4, x_1 : 0.3, x_2 : 0, x_3 : 0$. Taking the min over each orbit we have $k(Gx_0) = 0.3, k(Gx_2) = 0$. Taking the sum over $F = \{x_0, x_3\}$ we obtain $\mathrm{err}(h) \geq 0.3$.

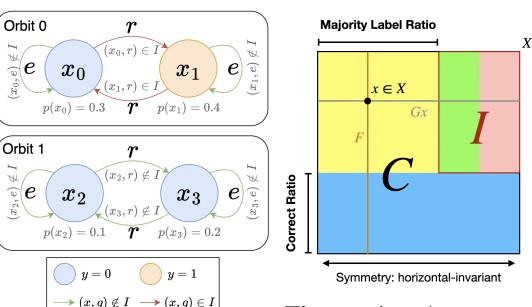

Figure 3: An example binary classification task. The circles are elements of $X$. The arrows show how $g \in G$ acts on $x \in X$. The arrow color shows whether $(x, g) \in I$.

Figure 4: An example multi-class classification task. Color indicates the label. The fundamental domain $F$ is a vertical line. For a point $x \in F$, the orbit $Gx$ is a horizontal line.

**Example 4.6** (Lower bound example for a multi-class classification task using Proposition 4.4)**.** Consider a multi-class classification task $f : \mathbb{R}^2 \to Y$ with $n = |Y|$ classes. For $x = (u, v) \in [0, 1]^2$ then $p(u, v) = 1$ and otherwise $p(u, v) = 0$; i.e., the support of $p$ is a unit square. Let $G$ denote the group of translations in the $u$-direction and $h$ a $G$-invariant network. In a data distribution illustrated in Figure 4, we compute the lower bound for $\mathrm{err}(h)$. Consider a fundamental domain $F$ (brown line in Figure 4). In the blue area, there is one label across the orbit (i.e., the horizontal

line), meaning $\forall g \in G, (x', g) \in C$, yielding Proposition 4.4 equals 0. For points in the yellow area, the majority label is yellow. This means that for $g \in G$ such that $gx$ is in yellow, $(x, g) \in C$; for other $g \in G, (x, g) \in I$. Consequently, Proposition 4.4 is equivalent to the combined green and pink lengths. Taking the integral over $F$ (Theorem 4.3), the lower bound equals the green and pink area ($I$ in Figure 4). We define correct ratio ($c$) as the blue area's height and majority label ratio ($m$) as the yellow area's length. Adjusting $c$ and $m$ transitions incorrect to correct equivariance, leading to $\text{err}(h) \geq area(I) = (1 - c) \times (1 - m)$. Appendix H.2 shows an experiment where the empirical result matches our analysis.

**Lower Bound When $G$ is Finite and The Action of $G$ is Density Preserving.** In this section, we consider the lower bound in Theorem 4.3 when $G$ is finite and the action of $G$ is density preserving, i.e., $p(gx) = p(x)$. Let $(Gx)_y = \{z \in Gx | f(z) = y\}$ be a subset of $Gx$ with label $y$. Define $\mathcal{Q}(x) = (\max_{y \in Y} |(Gx)_y|)/|Gx|$, which is the fraction of data in the orbit $Gx$ that has the majority label. Denote $Q = \{\mathcal{Q}(x) : x \in X\}$ the set of all possible values for $\mathcal{Q}$. Consider a partition of $X = \coprod_{q \in Q} X_q$ where $X_q = \{x \in X : \mathcal{Q}(x) = q\}$. Define $c_q = \mathbb{P}(x \in X_q) = |X_q|/|X|$.

**Proposition 4.7.** *The error lower bound $\text{err}(h) \geq 1 - \sum_q qc_q$ from Wang et al. [58] (Proposition 4.1) is a special case of Theorem 4.3.*

Proof in Appendix C. The proposition shows Theorem 4.3 is a strict generalization of [58, Prop 4.1].

## 4.2 Lower Bound for Invariant Regression

In this section, we give a lower bound of the error function $\text{err}_{\text{reg}}(h)$ (Equation 1) in a regression task given that $h$ is invariant, i.e., $h(gx) = h(x)$ for all $g \in G$. Assume $Y = \mathbb{R}^n$. Denote by $p(Gx) = \int_{z \in Gx} p(z)dz$ the probability of the orbit $Gx$. Denote by $q(z) = \frac{p(z)}{p(Gx)}$ the normalized probability density of the orbit $Gx$ such that $\int_{Gx} q(z)dz = 1$. Let $\mathbb{E}_{Gx}[f]$ be the mean of function $f$ on the orbit $Gx$ defined, and let $\mathbb{V}_{Gx}[f]$ be the variance of $f$ on the orbit $Gx$,

$$\mathbb{E}_{Gx}[f] = \int_{Gx} q(z)f(z)dz = \frac{\int_{Gx} p(z)f(z)dz}{\int_{Gx} p(z)dz}, \qquad \mathbb{V}_{Gx}[f] = \int_{Gx} q(x)||\mathbb{E}_{Gx}[f] - f(z)||_2^2.$$

**Theorem 4.8.** $\text{err}(h) \geq \int_F p(Gx)\mathbb{V}_{Gx}[f]dx.$

*Proof.* The error function (Equation 1) can be written as:

$$\text{err}(h) = \int_X p(x)||f(x) - h(x)||_2^2 dx = \int_{x \in F} \int_{z \in Gx} p(z)||f(z) - h(z)||_2^2 dzdx.$$

Denote $e(x) = \int_{Gx} p(z)||f(z) - h(z)||_2^2 dz$. Since $h$ is $G$-invariant, there exists $c \in \mathbb{R}^n$ such that $h(z) = c$ for all $z \in Gx$. Then $e(x)$ can be written as $e(x) = \int_{Gx} p(z)||f(z) - c||_2^2 dz$. Taking the derivative of $e(x)$ with respect to $c$ and setting it to 0 gives $c^*$, the minimum of $e(x)$, $c^* = \frac{\int_{Gx} p(z)f(z)dz}{\int_{Gx} p(z)dz} = \mathbb{E}_{Gx}[f]$. Substituting $c^*$ into $e(x)$ we have

$$e(x) \geq \int_{Gx} p(Gx)\frac{p(z)}{p(Gx)}||\mathbb{E}_{Gx}[f] - f(z)||_2^2 dz = p(Gx)\mathbb{V}_{Gx}[f].$$

We can obtain the claim by taking the integral of $e(x)$ over the fundamental domain $F$. $\qquad\square$

## 4.3 Lower Bound for Equivariant Regression

We now prove a lower bound for $\text{err}(h)$ in a regression task given the model $h$ is equivariant, that is, $h(\rho_X(g)x) = \rho_Y(g)h(x)$ where $g \in G, \rho_X$ and $\rho_Y$ are group representations associated with $X$ and $Y$. We will denote $\rho_X(g)x$ and $\rho_Y(g)y$ by $gx$ and $gy$, leaving the representation implicit. Assume $Y = \mathbb{R}^n$ and $\alpha(x, g)$ is the same as in equation 3. Let Id be the identity. Define a matrix $Q_{Gx} \in \mathbb{R}^{n \times n}$ and $q(gx) \in \mathbb{R}^{n \times n}$ so that $\int_G q(gx)dg = \text{Id}$ by

$$Q_{Gx} = \int_G p(gx)\rho_Y(g)^T\rho_Y(g)\alpha(x, g)dg, \qquad q(gx) = Q_{Gx}^{-1}p(gx)\rho_Y(g)^T\rho_Y(g)\alpha(x, g). \quad (5)$$

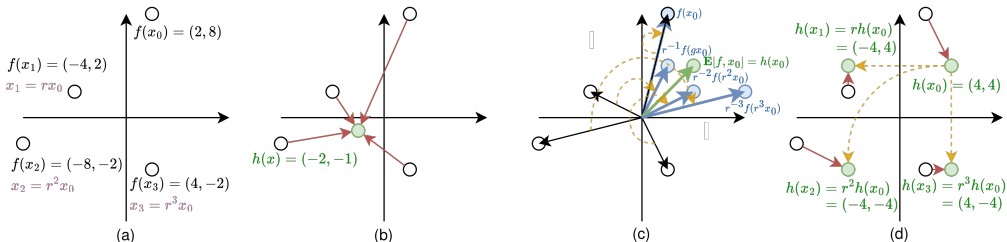

(a)         (b)         (c)         (d)

Figure 5: An example regression task. (a) The value of $f(x)$ and the transformation rule (purple) with respect to group $G = C_4$ for all $x \in X$. The four points belong to a single orbit. (b) When using an invariant network, the minimal error (red) is obtained when the invariant network outputs the mean value (green) of the orbit. (c) For an equivariant network, the minimizer (green) can be obtained by taking the mean of the $G$-stabilized $f(x)$ (inversely transformed) (blue) for all $x$ in the orbit with respect to the transformation rule in the orbit. (d) The minimal error of an equivariant network.

Here, for simplicity, we assume $Q_{Gx}$ is an invertible matrix. (See Appendix D for general case).

If $f$ is equivariant, $g^{-1}f(gx)$ is a constant for all $g \in G$. Define $\mathbf{E}_G[f, x]$

$$\mathbf{E}_G[f, x] = \int_G q(gx)g^{-1}f(gx)dg. \tag{6}$$

**Theorem 4.9.** *The error of $h$ has lower bound* $\mathrm{err}(h) \geq \int_F \int_G p(gx)\|f(gx) - g\mathbf{E}_G[f, x]\|_2^2 \alpha(x, g) dg dx.$

See Appendix D for the proof. Intuitively, $\mathbf{E}_G[f, x]$ is the minimizer obtained by taking the mean of all inversely transformed $f(x)$ for all $x$ in the orbit, see Figure 5cd and Example 4.11 below.

**Corollary 4.10.** *Denote $p(Gx) = \int_{Gx} p(z)dz$. Denote $q_x : g \mapsto q(gx)$. Define $G$-stabilized $f$ as $f_x : g \mapsto g^{-1}f(gx)$. When $\rho_Y$ is an orthogonal representation $\rho_Y : G \to \mathrm{O}(n) \subset GL(n)$, $q_x$ is a probability density function on $G$. Denote the variance of $f_x$ as $\mathbb{V}_G[f_x]$ where $g \sim q_x$. The error has a lower bound $\mathrm{err}(h) \geq \int_F p(Gx)\mathbb{V}_G[f_x]dx.$*

See Appendix E for the proof. Notice that Corollary 4.10 is a generalization of Theorem 4.8. That is, Theorem 4.8 can be recovered by taking $\rho_Y(g) = \mathrm{Id}$ (See the proof in Appendix F).

**Example 4.11** (Lower bound example of a regression task). Consider a regression problem where $X = \{x_0, x_1, x_2, x_3\}$ and $Y = \mathbb{R}^2$. Assume $p$ is uniform density. The cyclic group $G = C_4 = \{e, r, r^2, r^3\}$ (where $e = 0$ rotation and $r = \pi/2$ rotation) acts on $X$ through $x_1 = rx_0; x_2 = rx_1; x_3 = rx_2; x_0 = rx_3$ (i.e., there is only one orbit $Gx = X$). $g \in G$ acts on $y \in Y$ through $\rho_Y(g) = \left(\begin{smallmatrix} \cos g & -\sin g \\ \sin g & \cos g \end{smallmatrix}\right)$. Figure 5a shows the output of $f(x), \forall x \in X$. **First**, consider a $G$-invariant network $h$. Since there is only one orbit, Theorem 4.8 can be simplified as: $\mathrm{err}(h) \geq \mathbb{V}_X[f]$, the variance of $f$ over $X$. This can be calculated by first taking the mean of $f(x)$ then calculating the mean square error (MSE) from all $x$ to the mean (Figure 5b). **Second**, consider a $G$-equivariant network $h$. Since $G$ is discrete, $gx$ permutes the order of $X$, $\rho_Y$ is an orthogonal representation, and there is only one orbit, Corollary 4.10 can be written as $\mathrm{err}(h) \geq \mathbb{V}_G[f_x]$, the variance of $G$-stabilized $f$. First, to calculate $\mathbb{E}_G[f_x]$, let $x = x_0$, we stabilize $g$ from $f$ by $g^{-1}f(gx)$ for all $g \in G$, then take the mean (Figure 5c). We can then find $\mathbb{V}_G[f_x]$ by calculating the MSE between $f(x)$ and transformed mean $g\mathbb{E}_G[f_x]$ (Figure 5d). Appendix H.3 shows an experiment in this example's environment.

## 5   Harmful Extrinsic Equivariance

Wang et al. [58] demonstrate that extrinsic equivariance, where the symmetry imposed on the model leads to out-of-distribution data with respect to the input distribution, can lead to a higher performance on the original training data. In this section, we argue that this is not necessarily true in all cases, and there can exist scenarios where extrinsic equivariance can even be harmful to the learning problem.

Consider a binary classification task where the domain is discrete and contains only a set of four points $S \subset \mathbb{R}^3$, and their labels are either $\{-1, +1\}$ as shown in Figure 6a. We consider the probability density $p$ to be uniform for this domain, i.e., $p(x) = 1/4$ for the four points $S$, and $p = 0$ elsewhere.

This domain is used for both model training and testing so there is no distribution shift. We consider two model classes, $\mathcal{F}_N$, the set of all linear models, and $\mathcal{F}_E$, the set of all linear models which are invariant with respect to the cyclic group $C_2 = \{1, g\}$, where $g(x_1, x_2, x_3) = (x_1, x_2, -x_3)$. $\mathcal{F}_N$ corresponds to an unconstrained or a non-equivariant model class and $\mathcal{F}_E$ corresponds to an extrinsically equivariant class for this domain. For the labeling shown in Figure 6, the hyperplane $x_3 = 0$ correctly classifies all samples and is contained in $\mathcal{F}_N$. However, a function $f_e \in \mathcal{F}_E$ is equivalent to a linear classifier on $\mathbb{R}^2$ and effectively sees the data as Figure 6b[4]. This exclusive-or problem does not admit a linear solution (it can be correct for at most 3 points).

Concretely, we can compute the empirical Rademacher complexity, a standard measure of model class expressivity, for non-equivariant and extrinsically equivariant model classes and show that $\mathcal{F}_E$ has lower complexity than $\mathcal{F}_N$. Recall that empirical Rademacher complexity is defined as $\mathfrak{R}_S(\mathcal{F}) = \mathbb{E}_\sigma \left[ \sup_{f \in \mathcal{F}} \frac{1}{m} \sum_{i=1}^m \sigma_i f(x^i) \right]$, where $S$ is the set of $m$ samples and $\sigma = (\sigma_1, \ldots, \sigma_m)^\top, \sigma_i \in \{-1, +1\}$ are independent uniform Rademacher random variables, and $x^i$ is the $i$-th sample. As there exists some linear function $f_n \in \mathcal{F}_N$ that fully classifies $S$ for any combination of labels, $\mathfrak{R}_S(\mathcal{F}_N) = 1$. For the extrinsic equivariance case, of the 16 possible label combinations, there are two cases where $f_e \in \mathcal{F}_E$ can at most classify 3 out of 4 points correctly, and thus $\mathfrak{R}_S(\mathcal{F}_E) = \frac{31}{32} < \mathfrak{R}_S(\mathcal{F}_N)$ (see Appendix G for the calculations). This illustrates that in certain cases, extrinsic equivariance can lead to lower model expressivity than no equivariance and thus be harmful to learning.

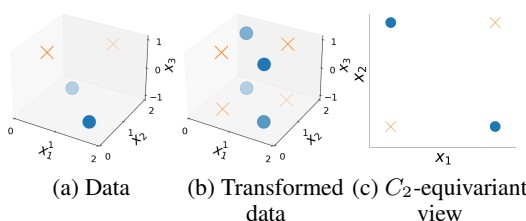

(a) Data    (b) Transformed data    (c) $C_2$-equivariant view

Figure 6: An example dataset where extrinsic equivariance increases the problem difficulty. The samples are of the form $x = (x_1, x_2, x_3)$ and the labels are shown as different shapes. A $C_2$-equivariant linear model transforms the original data (a) into (b), which is equivalent to viewing the data as in (c). The original task has an easy solution (e.g. hyperplane at $x_3 = 0$), while the $C_2$-invariant view is the classic exclusive-or problem.

## 6 Experiments

We perform experiments to validate our theoretical analysis on both the lower bounds (Section 4) and the harmful extrinsic equivariance (Section 5). We find that our bounds accurately predict empirical model error. In addition to the experiments in this section, Appendix H.2 shows an experiment verifying our classification bound (Theorem 4.3) and Appendix H.3 shows an experiment verifying our regression bound (Theorem 4.8 and 4.9). The experiment details are in Appendix I.

### 6.1 Swiss Roll Experiment

We first perform an experiment in a vertically separated Swiss Roll data distribution, see Figure 7a[5]. This example, similar to that in Section 5, demonstrates that a $C_2$-invariant model effectively "flattens" the $z$-dimension of the data so it must learn the decision boundary between two spirals (Figure 7b), whereas the non-equivariant model only needs to learn a horizontal plane to separate the classes, a significantly easier task. Besides the extrinsic data distribution, we consider two other data distributions shown in Figure 7c and Figure 7d, where a $C_2$-invariant model will observe incorrect and correct equivariance due to the mismatched and matched data labels in the two $z$ planes.

We combine data from all three distributions in various proportions to test the performance of a $z$-invariant network (INV) with a baseline unconstrained network (MLP). Let $c$ be the correct ratio, the proportion of data from the correct distribution. Define the incorrect ratio $i$ and extrinsic ratio $e$ similarly. We consider all $c, i, e$ that are multiples of 0.125 such that $c + i + e = 1$. Figure 7ef shows some example data distributions. Relative to INV, this mixed data distribution has partial correct, incorrect, and extrinsic equivariance, which is not fully captured in prior work [58]. Based on Proposition 4.4, we have $k(Gx) = 0.5$ for $x$ drawn from the incorrect distribution, and $k(Gx) = 0$ otherwise. Since the data is evenly distributed, we can calculate the error lower bound $\text{err}(h) \geq 0.5i$.

---

[4]Notice that the four additional points in Figure 6b compared with (a) are not in the domain, they are created through applying the transformation rule of $\mathcal{F}_E$ onto the domain.

[5]For visualization, we show a simpler version of the data distribution. See Appendix H.1 for the actual one.

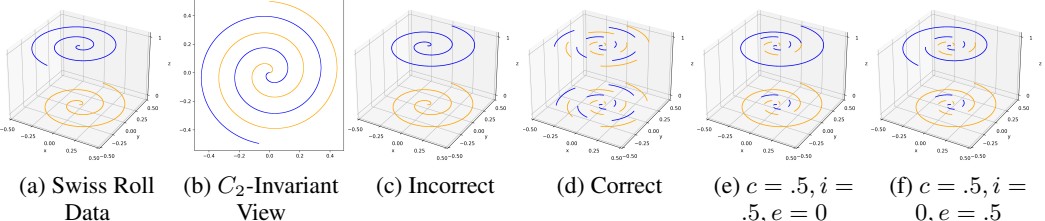

(a) Swiss Roll Data    (b) $C_2$-Invariant View    (c) Incorrect    (d) Correct    (e) $c = .5, i = .5, e = 0$    (f) $c = .5, i = 0, e = .5$

Figure 7: (a) (b) The Swiss Roll data distribution that leads to harmful extrinsic equivariance. (c) (d) The correct and incorrect data distribution in the Swiss Roll experiment. Here the spirals overlap with mismatched and matched labels respectively. (e) (f) Data distribution example with different correct ratio ($c$), incorrect ratio ($i$), and extrinsic ratio ($e$) values.

**Results.** Figure 8a shows the test success rate of INV compared with MLP when $e$ and $c$ vary with $i = 0$. When $e$ increases, the performance of INV decreases while the performance of MLP shows an inverse trend, demonstrating that extrinsic equivariance is harmful in this experiment. Figure 8b shows the performance of INV and MLP when $c$ and $i$ vary while $e = 0$. The green line shows the upper bound of the test success rate ($1 - 0.5i$). The experimental result matches our theoretical analysis quite closely. Notice that when $c$ increases, there is a bigger gap between the performance of the network and its theoretical upper bound, since classification in the correct distribution is a harder task. Appendix H.1 shows the complete results of this experiment.

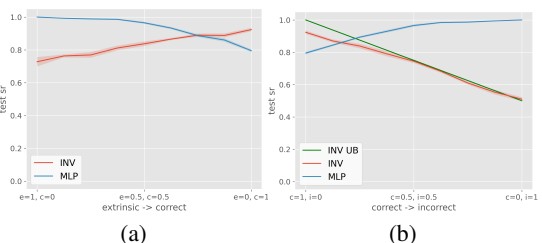

Figure 8: Result of the Swiss Roll experiment. (a) test success rate of an invariant network (red) and an unconstrained MLP (blue) with different extrinsic and correct ratio when incorrect ratio is 0. (b) same as (a) with different correct and incorrect ratio when extrinsic ratio is 0. Averaged over 10 runs.

## 6.2 Digit Classification Experiment

In this experiment, we apply our theoretical analysis to a realistic digit classification task using both the printed digit dataset [16] and the MNIST handwritten digit dataset [15]. We compare a $D_4$-invariant network ($D_4$) with an unconstrained CNN. In the printed digit classification, $D_4$ exhibits incorrect equivariance for 6 and 9 under a $\pi$ rotation. Using Theorem 4.3, we can calculate a lower bound of error for $D_4$ at 10%. However, as shown in Table 1 (top), the experimental results indicate that the actual performance is slightly better than predicted by the theory. We hypothesize that this discrepancy arises because a rotated 9 differs slightly from a 6 in some fonts. We conduct a similar experiment using the MNIST handwritten digit dataset (Table 1 bottom), where $D_4$ achieves even better performance in classifying 6 and 9. This improvement is likely due to the more distinguishable handwriting of these digits, although the performance still underperforms the CNN as incorrect equivariance persists. It is important to note that there is a significant decrease in performance for $D_4$ when classifying 2/5 and 4/7 compared to the CNN. This is because a vertical flip results in incorrect equivariance when classifying handwritten 2/5, and a similar issue arises for 4/7 under a $\pi/2$ rotation followed by a vertical flip (notice that Weiler and Cesa [60] make a similar observation). These experiments demonstrate that our theory is useful not only for calculating the performance bounds of an equivariant network beforehand, but also for explaining the suboptimal performance of an equivariant network, thereby potentially assisting in model selection.

## 6.3 Robotic Experiment

In this experiment, we evaluate our theory in behavior cloning in robotic manipulation. We first preform an experiment where the problem is a mixture of correct and incorrect equivariance for a $D_1$-equivariant policy network ($D_1$) where the robot's action will flip when the state is flipped

| Digit | Overall | 0 | 1 | 2 | 3 | 4 | 5 | 6 | 7 | 8 | 9 |
|---|---|---|---|---|---|---|---|---|---|---|---|
| Print CNN | 96.62 | 99.81 | 98.27 | 97.06 | 96.12 | 98.04 | 92.9 | 94.93 | 97.85 | 93.65 | 96.13 |
| Print D4 | 92.5 | 99.71 | 98.62 | 96.94 | 95.98 | 97.91 | 93.52 | **63.08** | 98.42 | 95.88 | **76.17** |
| Print D4 Upper Bound | 90 | 100 | 100 | 100 | 100 | 100 | 100 | 50 | 100 | 100 | 50 |
| MNIST CNN | 98.21 | 99.51 | 99.61 | 98.62 | 98.83 | 98.08 | 98.47 | 97.99 | 97.04 | 96.98 | 96.81 |
| MNIST D4 | 96.15 | 98.93 | 99.21 | **91.84** | 98.28 | **95.49** | **95.04** | **93.71** | **95.67** | 97.73 | **95.34** |

Table 1: $D_4$-invariant network compared with an unconstrained CNN in printed and MNIST handwritten digit classification tasks. Bold indicates that there is a $> 1\%$ difference in two models.

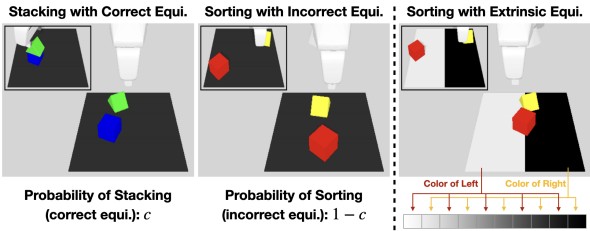

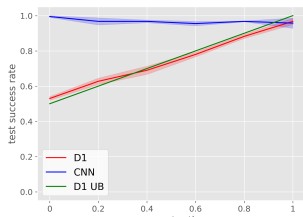

Figure 9: Left: an environment containing both correct (Stacking) and incorrect (Pushing) equivariance for a $D_1$-equivariant (horizontal flip) policy net. Right: an environment with harmful extrinsic equivariance for the same policy.

Figure 10: Result of robotic experiment with different correct ratio. Averaged over 4 runs.

horizontally. Specifically, the environment contains two possible tasks (Figure 9 left). Stacking requires the robot to stack a green triangle on top of a blue cube. Here flip equivariance is correct. Sorting requires the robot to push the red cube to the left and the yellow triangle to the right. Here flip equivariance is incorrect because the robot should not sort the objects in an opposite way when the state is flipped (in other words, $D_1$ cannot distinguish left and right). We vary the probability $c$ of the stacking task (correct ratio) in the task distribution, and compare the performance of $D_1$ versus a standard CNN policy. If we view the sorting task as a binary classification task, we can calculate an upper bound of performance for $D_1$ using Theorem 4.3: $0.5 + 0.5c$. Figure 10 shows the result. Notice that the performance of $D_1$ closely matches the theoretical upper bound, while the performance of CNN remains relatively stable for all Stacking-Sorting distributions.

We further evaluate $D_1$ in a sorting task with harmful extrinsic equivariance. Here, the goal for the robot is the same as sorting above (push the red cube left and the yellow triangle right), however, the left and right sides of the workspace can now be differentiated by gray-scale colors. The shades of gray are discretized evenly into $n$ bins, where the left side's color is randomly sampled from the odd-numbered bins, and the right side's color is randomly sampled from the even-numbered bins (Figure 9 right). The different color distributions of the left and right sides make $D_1$ extrinsically equivariant, but it needs to learn the color distribution to distinguish left and right (while CNN can distinguish left and right directly). We set $n = 10$, and $D_1$ achieves $71.5 \pm 1.6\%$ test success rate, while CNN achieves $99.5 \pm 0.5\%$, demonstrating that the $D_1$ extrinsic equivariance is harmful in this task. See Appendix I.5 for the details of the robot experiment.

## 7  Discussion

This paper presents a general theory for when the symmetry of the ground truth function and equivariant network are mismatched. We define pointwise correct, incorrect, and extrinsic equivariance, generalizing prior work [58] to include continuous mixtures of the three extremes. We prove error lower bounds for equivariant networks applied to asymmetric tasks including classification, invariant regression, and equivariant regression without the assumption of invariant data density. Our work discusses the potential disadvantage of extrinsic equivariance, and provides experiments that validate our theoretical analysis. The major limitation of this paper is that our theoretical lower bounds require domain knowledge like the density function over the domain. In future work, we will develop easy-to-apply model selection tools using our theory. Another future direction is theoretically understanding when extrinsic equivariance is helpful or harmful and analyzing the effect of extrinsic equivariance on the decision boundary of an equivariant network.

## Acknowledgments

This work is supported in part by NSF 1724257, NSF 1724191, NSF 1763878, NSF 1750649, NSF 2107256, NSF 2134178, NSF 2312171, and NASA 80NSSC19K1474.

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

## A   Integrals on the Group

**Fundamental Domains**   In this paper, we are interested in cases in which the group $G$ is not necessarily discrete but may have positive dimension. We do not assume the fundamental domain has non-empty interior, and thus domain is a misnomer. In this case the conjugates of the fundamental domain $gF$ have measure 0 and the condition that their intersection have measure 0 is vacuous. Instead we assume a stronger condition, that the union of all pairwise intersections $\bigcup_{g_1 \neq g_2} (g_1 F \cap g_2 F)$ has measure 0. We also require that $F$ and the orbits $Gx$ are differentiable manifolds such that integrals over $X$ may be evaluated $\int_X f(x)dx = \int_F \int_{Gy} f(z)dzdy$ similar to Equation 8 from [18].

**Reparameterization**   Consider the integral

$$\int_{Gx} f(z)dz. \tag{7}$$

Denote the identification of the orbit $Gx$ and coset space $G/G_x$ with respect to the stabilizer $G_x = \{g : gx = x\}$ by $a_x \colon G/G_x \to Gx$. Then the integral can be written

$$\int_{G/G_x} f(\bar{g}x) \left| \frac{\partial a_x(\bar{g})}{\partial \bar{g}} \right| d\bar{g}.$$

We can also lift the integral to $G$ itself

$$\int_{G/G_x} f(\bar{g}x) \left| \frac{\partial a_x(\bar{g})}{\partial \bar{g}} \right| d\bar{g} = \left( \int_{G_x} dh \right)^{-1} \left( \int_{G_x} dh \right) \int_{G/G_x} f(\bar{g}x) \left| \frac{\partial a_x(\bar{g})}{\partial \bar{g}} \right| d\bar{g}$$

$$= \left( \int_{G_x} dh \right)^{-1} \int_{G/G_x} \int_{G_x} f(\bar{g}hx) \left| \frac{\partial a_x(\bar{g})}{\partial \bar{g}} \right| dhd\bar{g}$$

$$= \left( \int_{G_x} dh \right)^{-1} \int_{G} f(gx) \left| \frac{\partial a_x(\bar{g})}{\partial \bar{g}} \right| dg.$$

Define $\alpha(g, x) = \left( \int_{G_x} dh \right)^{-1} \left| \frac{\partial a_x(\bar{g})}{\partial \bar{g}} \right|$. Then

$$\int_{Gx} f(z)dz = \int_G f(gx)\alpha(g, x)dg.$$

## B   Iterated Integral

Let $X$ be an $n$-dimensional space, Definition 4.2 (Equation 2) defines $k(Gx)$ as an integral over $Gx \subseteq X$, which is a $m$-dimensional sub-manifold of $X$. In Theorem 4.3, Equation 4 rewrites the error function (Equation 1) as an iterated integral over the orbit $Gx$ and then the fundamental domain $F$ using Definition 4.1. In the discrete group case, $m$ would be 0, Equation 2 is an integral of a 0-form in a 0-manifold, which is a sum:

$$k(Gx) = \min_{y \in Y} \sum_{z \in Gx} p(z)\mathbb{1}(f(z) \neq y) = \min_{y \in Y} \sum_{g \in G} p(gx)\mathbb{1}(f(gx) \neq y) \tag{8}$$

## C   Proof of Proposition 4.7

*Proof.* Consider the integral of probability density inside $Gx$, for a given $y$, it can be separated into two groups:

$$\int_{Gx} p(z)dz = \int_{Gx} p(z)\mathbb{1}(f(z) = y)dz$$

$$+ \int_{Gx} p(z)\mathbb{1}(f(z) \neq y)dz.$$

We can then rewrite $k(Gx)$ in Equation 2 as:

$$k(Gx) = \min_{y \in Y} \left[ \int_{Gx} p(z)dz - \int_{Gx} p(z)\mathbb{1}(f(z) = y)dz \right]. \tag{9}$$

Letting $(Gx)_y = \{x' \in Gx \mid f(x') = y\} = f^{-1}(y) \cap Gx$, Equation 9 can be written as:

$$k(Gx) = \min_{y \in Y} \left[ \int_{Gx} p(z)dz - \int_{(Gx)_y} p(z)dz \right]$$

$$\int_{Gx} p(z)dz - \max_{y \in Y} \int_{(Gx)_y} p(z)dz.$$

Theorem 4.3 can be rewritten as:

$$\text{err}(h) \geq \int_F \left( \int_{Gx} p(z)dz - \max_{y \in Y} \int_{(Gx)_y} p(z)dz \right) dx$$

$$\geq \int_F \int_{Gx} p(z)dz - \int_F \max_{y \in Y} \int_{(Gx)_y} p(z)dz$$

$$\geq 1 - \int_F \max_{y \in Y} |(Gx)_y| p(x)dx. \tag{10}$$

The first term in Equation 10 uses the fact that $X = \bigcup_{x \in F} Gx$ so the integral of the probability of the orbits of all points in the fundamental domain is the integral of the probability of the input domain $X$ which is 1. The second term of Equation 10 uses $p(gx) = p(x)$ so the integration of $p(z)$ on $(Gx)_y$ becomes $p(x)$ times the range of the limit which is the size of $(Gx)_y$, $|(Gx)_y|$.

Now consider a partition of $F = \coprod_q F_q$ where $F_q = \{x \in F : (\max_{y \in Y} |(Gx)_y|)/|Gx| = q\}$. We can rewrite Equation 10 as:

$$\text{err}(h) \geq 1 - \int_F q|Gx|p(x)dx \tag{11}$$

$$\geq 1 - \sum_q \int_{F_q} q|Gx|p(x)dx \tag{12}$$

$$\geq 1 - \sum_q q \int_{F_q} |Gx|p(x)dx. \tag{13}$$

Equation 11 uses the definition of $q$. Equation 12 separates the integral over $F$ into the partition of $F$. Equation 13 moves $q$ out from the integral because it is a constant inside the integral. Consider the definition of $c_q$, we have:

$$c_q = \mathbb{P}(x \in X_q)$$

$$= \int_{X_q} p(x)dx$$

$$= \int_{F_q} \int_{Gx} p(z)dzdx \tag{14}$$

$$= \int_{F_q} |Gx|p(x)dx. \tag{15}$$

Equation 14 uses $X_q = \bigcup_{x \in F_q} Gx$. Equation 15 uses $p(x) = p(gx)$. Now we can write Equation 13 as:

$$\text{err}(h) \geq 1 - \sum_q qc_q.$$

$\square$

# D   Proof of Theorem 4.9

Define $q(gx) \in \mathbb{R}^{n \times n}$ such that

$$Q_{Gx}q(gx) = p(gx)\rho_Y(g)^T\rho_Y(g)\alpha(x, g). \tag{16}$$

In particular, $q(gx)$ exists when $Q_{Gx}$ is full rank. It follows that $Q_{Gx}\int_G q(gx)dg = Q_{Gx}$. Moreover, $Q_{Gx}$ and $q(gx)$ are symmetric matrix.

*Proof.* The error function (Equation 1) can be written

$$
\begin{aligned}
\text{err}(h) &= \mathbb{E}_{x \sim p}[||f(x) - h(x)||_2^2] \\
&= \int_X p(x)||f(x) - h(x)||_2^2 dx \\
&= \int_{x \in F} \int_{g \in G} p(gx)||f(gx) - h(gx)||_2^2 \alpha(x, g)dgdx.
\end{aligned}
$$

Denote $e(x) = \int_G p(gx)||f(gx) - h(gx)||_2^2\alpha(x, g)dg$. Since $h$ is $G$-equivariant, for each $x \in F$ the value $c = h(x) \in \mathbb{R}^n$ of $h$ at $x$ determines the value of $h$ across the whole orbit $h(gx) = gh(x) = gc$ for $g \in G$. Then $e(x)$ can be written

$$
\begin{aligned}
e(x) &= \int_G p(gx)||f(gx) - gc||_2^2\alpha(x, g)dg \\
&= \int_G p(gx)||g(g^{-1}f(gx) - c)||_2^2\alpha(x, g)dg \\
&= \int_G (g^{-1}f(gx) - c)^T p(gx)g^T g\alpha(x, g)(g^{-1}f(gx) - c)dg \\
&= \int_G (g^{-1}f(gx) - c)^T Q_{Gx}q(gx)(g^{-1}f(gx) - c)dg. \tag{17}
\end{aligned}
$$

Taking the derivative of $e(x)$ with respect to $c$ we have

$$
\begin{aligned}
\frac{\partial e(x)}{\partial c} &= \int_G \left((Q_{Gx}q(gx))^T + (Q_{Gx}q(gx))\right)(c - g^{-1}f(gx))dg \\
&= \int_G 2Q_{Gx}q(gx)(c - g^{-1}f(gx))dg.
\end{aligned}
$$

Setting $\partial e(x)/\partial c = 0$ we can find an equation for $c^*$ which minimizes $e(x)$

$$
\begin{aligned}
Q_{Gx}\int_G q(gx)dg \cdot c^* &= Q_{Gx}\int_G q(gx)g^{-1}f(gx)dg \\
Q_{Gx}c^* &= Q_{Gx}\mathbb{E}_G[f, x]. \tag{18}
\end{aligned}
$$

Substituting $c^*$ into Equation 17 we have

$$
\begin{aligned}
e(x) &\geq \int_G (g^{-1}f(gx) - c^*)^T Q_{Gx}q(gx)(g^{-1}f(gx) - c^*)dg \\
&= \int_G (g^{-1}f(gx))^T Q_{Gx}q(gx)(g^{-1}f(gx)) \\
&\quad - \left(c^{*T}Q_{Gx}q(gx)g^{-1}f(gx)\right)^T \\
&\quad - c^{*T}Q_{Gx}q(gx)g^{-1}f(gx) \\
&\quad + c^{*T}Q_{Gx}q(gx)c^*dg. \tag{19}
\end{aligned}
$$

The term $\int_G c^{*T}Q_{Gx}q(gx)g^{-1}f(gx)dg$ could be simplified as

$$\int_G c^{*T} Q_{Gx} q(gx) g^{-1} f(gx) dg = \int_G \mathbf{E}_G[f, x] Q_{Gx} q(gx) g^{-1} f(gx) dg.$$

$$(20)$$

Notice that $Q_{Gx}$, and $q(gx)$ are symmetric matrix

$$\int_G c^{*T} Q_{Gx} q(gx) c^* dg = \int_G c^{*T} q(gx) Q_{Gx} c^* dg$$
$$= \int_G \mathbf{E}_G^T[f, x] Q_{Gx} q(gx) \mathbf{E}_G[f, x] dg.$$

Thus Equation 19 becomes

$$\begin{aligned}
e(x) \geq & \int_G (g^{-1} f(gx))^T Q_{Gx} q(gx) (g^{-1} f(gx)) \\
& - \left( \mathbf{E}_G^T[f, x] Q_{Gx} q(gx) g^{-1} f(gx) \right)^T \\
& - \mathbf{E}_G^T[f, x] Q_{Gx} q(gx) g^{-1} f(gx) \\
& + \mathbf{E}_G^T[f, x] Q_{Gx} q(gx) \mathbf{E}_G[f, x] dg \\
= & \int_G p(gx) ||f(gx) - g\mathbf{E}_G[f, x]||_2^2 \alpha(x, g) dg.
\end{aligned}$$

Taking the integral over the fundamental domain $F$ we have

$$\begin{aligned}
\text{err}(h) &= \int_F e(x) \\
&\geq \int_F \int_G p(gx) ||f(gx) - g\mathbf{E}_G[f, x]||_2^2 \alpha(x, g) dg dx.
\end{aligned}$$

$$(21)$$

$\square$

# E    Proof of Corollary 4.10

*Proof.* When $\rho_Y$ is an orthogonal representation, we have $\rho_Y(g)^T \rho_Y(g) = I_n$, i.e., the identity matrix. Then $q(gx)$ can be written as $q(gx) = s(gx)\text{Id}$ where $s(gx)$ is a scalar. Since $\int_G q(gx) dg = \text{Id}$, we can re-define $q(gx)$ to drop Id and only keep the scalar, then $q_x(g)$ can be viewed as a probability density function of $g$ because now $\int_G q_x(g) = 1$.

With $q_x(g)$ being the probability density function, $\mathbf{E}_G[f, x]$ (Equation 6) naturally becomes the mean $\mathbb{E}_G[f_x]$ where $g \sim q_x$.

Now consider $e(x) = \int_G p(gx) ||f(gx) - g\mathbf{E}_G[f_x]||_2^2 \alpha(x, g) dg$ in Theorem 4.9, it can be written as

$$\begin{aligned}
e(x) &= \int_G p(gx) ||f(gx) - g\mathbb{E}_G[f_x]||_2^2 \alpha(x, g) dg \\
&= \int_G p(gx) ||g(g^{-1} f(gx) - \mathbb{E}_G[f_x])||_2^2 \alpha(x, g) dg \\
&= \int_G p(gx) (g^{-1} f(gx) - \mathbb{E}_G[f_x])^T \rho_Y(g)^T \rho_Y(g) (g^{-1} f(gx) - \mathbb{E}_G[f_x]) \alpha(x, g) dg.
\end{aligned}$$

Since $\rho_Y(g)^T \rho_Y(g) = I_n$, we have

$$\begin{aligned}
e(x) &= \int_G p(gx) (g^{-1} f(gx) - \mathbb{E}_G[f_x])^T (g^{-1} f(gx) - \mathbb{E}_G[f_x]) \alpha(x, g) dg \\
&= \int_G p(gx) ||g^{-1} f(gx) - \mathbb{E}_G[f_x]||_2^2 \alpha(x, g) dg.
\end{aligned}$$

$$(22)$$

From Equation 5 we have $p(gx)\alpha(x,g) = Q_{Gx}q(gx)$. Substituting in Equation 22 we have

$$e(x) = \int_G Q_{Gx}q(gx)||g^{-1}f(gx) - \mathbb{E}_G[f_x]||_2^2 dg.$$

Since $Q_{Gx} = \int_G p(gx)\alpha(a,g)dg$ when $\rho_Y(g)^T \rho_Y(g) = I_n$, we have

$$\begin{aligned} e(x) =& Q_{Gx}\int_G q_x(g)||g^{-1}f(gx) - \mathbb{E}_G[f_x]||_2^2 dg \\ =& Q_{Gx}\mathbb{V}_G[f_x]. \end{aligned} \tag{23}$$

Now consider $Q_{Gx}$ (Equation 5), when $\rho_Y(g)^T \rho_Y(g) = I_n$, it can be written

$$\begin{aligned} Q_{Gx} =& \int_G p(gx)\alpha(x,g)dg \\ =& \int_{Gx} p(z)dz \\ =& p(Gx). \end{aligned}$$

Replacing $Q_{Gx}$ with $p(Gx)$ in Equation 23 then taking the integral of $e(x)$ over the fundamental domain gives the result. $\qquad\square$

# F  Lower Bound of Equivariant Regression when $\rho_Y = \mathrm{Id}$

**Proposition F.1.** *When $\rho_Y = \mathrm{Id}$, the error of $h$ has lower bound $\mathrm{err}(h) \geq \int_F p(Gx)\mathbb{V}_{Gx}[f]dx$, which is the same as Theorem 4.8.*

*Proof.* Consider Equation 5, when $\rho_Y(g) = \mathrm{Id}$, we have

$$Q_{Gx} = \int_G p(gx)\alpha(x,g)dg.$$

Exchange the integration variable using $z = gx$ we have

$$Q_{Gx} = \int_{Gx} p(z)dz. \tag{24}$$

Consider $\mathbb{E}_G[f_x] = \int_G q_x(g)g^{-1}f(gx)dg$. When $\rho_Y(g) = \mathrm{Id}$, it becomes

$$\mathbb{E}_G[f_x] = \int_G q(gx)f(gx)dg.$$

Substituting $q(gx)$ with Equation 5, considering $\rho_Y(g) = \mathrm{Id}$, we have

$$\mathbb{E}_G[f_x] = \int_G Q_{Gx}^{-1}p(gx)f(gx)\alpha(x,g)dg.$$

Exchange the integration variable using $z = gx$ we have

$$\mathbb{E}_G[f_x] = \int_{Gx} Q_{Gx}^{-1}p(z)f(z)dz.$$

Substituting Equation 24 we have

$$\begin{aligned} \mathbb{E}_G[f_x] =& \int_{Gx} \frac{p(z)}{\int_{Gx} p(z)dz}f(z)dz \\ =& \mathbb{E}_{Gx}[f]. \end{aligned}$$

Similarly, we can proof $\mathbb{V}_G[f_x] = \mathbb{V}_{Gx}[f]$, thus when $\rho_Y = \mathrm{Id}$, Corollary 4.10 is Theorem 4.8. $\quad\square$

| | Non-equivariant model class | $C_2$-equivariant model class |
| --- | --- | --- |
| $\sigma^\top$ | $\sup_{f_n \in \mathcal{F}_N} \frac{1}{m} \sum_{i=1}^{m} \sigma_i f_n(x^i)$ | $\sup_{f_n \in \mathcal{F}_N} \frac{1}{m} \sum_{i=1}^{m} \sigma_i f_n(x^i)$ |
| $[-1, -1, -1, -1]$ | 1 | 1 |
| $[-1, -1, -1, +1]$ | 1 | 1 |
| $[-1, -1, +1, -1]$ | 1 | 1 |
| $[-1, -1, +1, +1]$ | 1 | 0.75 |
| $[-1, +1, -1, -1]$ | 1 | 1 |
| $[-1, +1, -1, +1]$ | 1 | 1 |
| $[-1, +1, +1, -1]$ | 1 | 1 |
| $[-1, +1, +1, +1]$ | 1 | 1 |
| $[+1, -1, -1, -1]$ | 1 | 1 |
| $[+1, -1, -1, +1]$ | 1 | 1 |
| $[+1, -1, +1, -1]$ | 1 | 1 |
| $[+1, -1, +1, +1]$ | 1 | 1 |
| $[+1, +1, -1, -1]$ | 1 | 0.75 |
| $[+1, +1, -1, +1]$ | 1 | 1 |
| $[+1, +1, -1, +1]$ | 1 | 1 |
| $[+1, +1, +1, +1]$ | 1 | 1 |
| $\mathfrak{R}_S$ | 1 | $\frac{31}{32}$ |

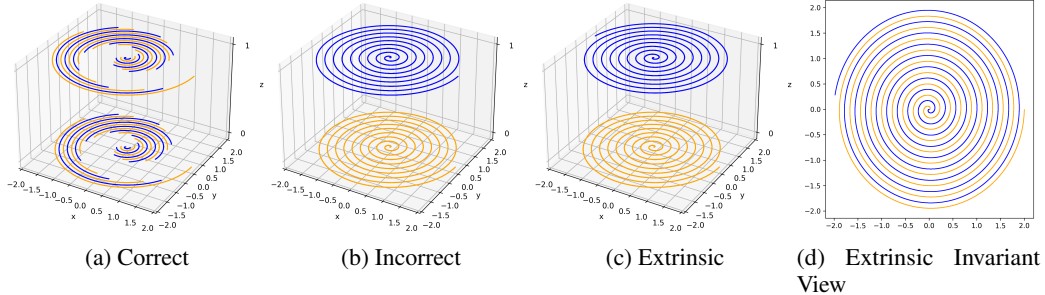

(a) Correct  (b) Incorrect  (c) Extrinsic  (d) Extrinsic Invariant View

Figure 11: The correct, incorrect, and extrinsic data distribution in the Swiss Roll experiment.

## G  Rademacher Complexity of Harmful Extrinsic Equivariance Example

Let $S = \{x^1, x^2, x^3, x^4\}$, where the labels are $y^1, y^2 = +1$ and $y^3, y^4 = -1$. We consider two model classes $\mathcal{F}_N$, the set of all linear models, and $\mathcal{F}_E$, the set of all linear models equivariant to $C_2$, and compute their empirical Rademacher complexity on $S$.

For the data $S$, an extrinsically equivariant linear model class has lower empirical Rademacher complexity than its unconstrained linear counterpart, demonstrating that extrinsic equivariance can be harmful to learning.

## H  Additional Experiments

### H.1  Swiss Roll Experiment

Figure 11 and Figure 12 show the actual data distribution for the Swiss Roll experiment in Section 6.1. In the incorrect distribution, the data in the two $z$ planes form two spirals with different labels but the same shape. The equivariance is incorrect because if we translate one spiral to the other spiral's plane, they will overlap but their labels are different. In the correct distribution, there are two different 'dashed' spirals copied into two $z$-planes. The equivariance is correct because after a $z$-translation, both the data and their labels exactly overlap. In all three cases, we assume the data has a uniform distribution. Figure 13b shows the ternary plot of MLP for all different $c, ir, er$, where the performance of MLP decreases as the correct ratio increases. Figure 13a shows an inverse trend: the

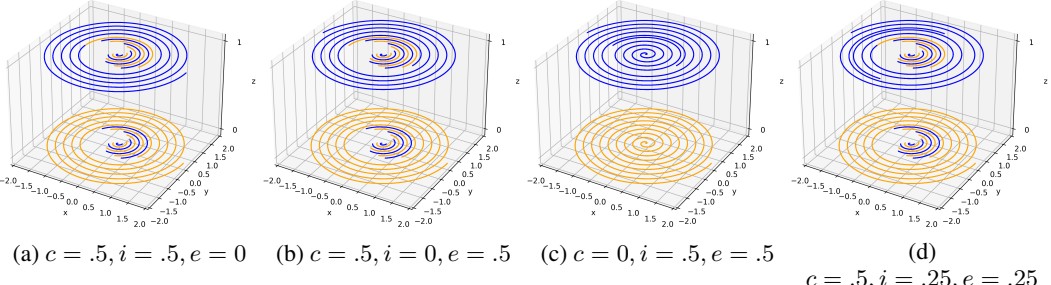

(a) $c = .5, i = .5, e = 0$    (b) $c = .5, i = 0, e = .5$    (c) $c = 0, i = .5, e = .5$      (d)
$c = .5, i = .25, e = .25$

Figure 12: Data distribution example with different correct ratio ($c$), incorrect ratio ($ir$), and extrinsic ratio ($er$) values.

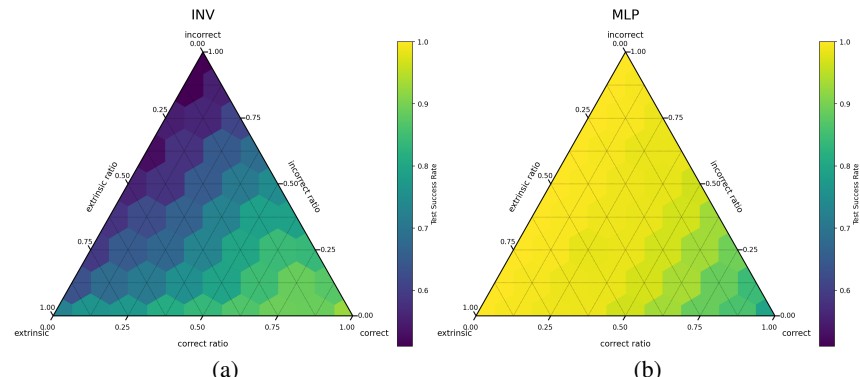

Figure 13: The ternary plot of the invariant network (a) and unconstrained network (b) with different correct, incorrect, and extrinsic ratio.

performance of INV increases as the correct ratio increases. Moreover, both extrinsic and incorrect equivariance harms the performance of INV, but incorrect equivariance is more devastating because the error is limited by a theoretical lower bound.

## H.2 Square Experiment

We consider the environment shown in Example 4.6. We vary $m \in \{0.2, 0.4, 0.6, 0.8, 1\}$ and $c \in \{0, 0.2, 0.4, 0.6, 0.8, 1\}$. We train an $u$-invariant network and evaluate its test performance with the theoretical lower bound $\mathrm{err}(h) \geq (1 - c) \times (1 - m)$. Figure 14 shows the test error of the trained network compared with the theoretical lower bound. The highest difference is below 3%, demonstrating the correctness of our theory.

## H.3 Regression Experiment

In this experiment, we validate our theoretical error lower bound for invariant and equivariant regression (Theorem 4.8, 4.9) in an environment similar to Example 4.11. Consider a regression task $f : \mathbb{R} \times \mathcal{X} \to \mathbb{R}^2$ given by $(\theta, x) \mapsto y$, where $\mathcal{X} = \{x_0, x_1, x_2, x_3\}$. The group $g \in G = C_4 = \{e, r, r^2, r^3\}$ acts on $(\theta, x)$ by $g(\theta, x) = (\theta, gx)$ through permutation: $x_1 = rx_0; x_2 = rx_1; x_3 =$

| | Invariant Network | Equivariant Network |
|---|---|---|
| Empirical/Theoretical | $1.002 \pm 0.000$ | $1.001 \pm 0.000$ |

Table 2: Empirical $\mathrm{err}(h)$ divided by theoretical $\mathrm{err}(h)$ for invariant regression and equivariant regression. Results are averaged over 100 runs with different $f$ for each regression. Empirical regression error matches theoretical error.

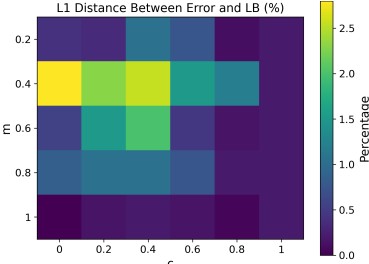

Figure 14: Result of the square experiment in terms of the $L_1$ distance between the network error and the theoretical lower bound in percentage. Each cell corresponds to an experiment with a particular correct ratio ($c$) and majority label ratio ($m$). Results are averaged over 10 runs.

$rx_2; x_0 = rx_3$. Let $r^k \in G$ acts on $y$ by $\rho_Y(g) = \begin{pmatrix} \cos g & -\sin g \\ \sin g & \cos g \end{pmatrix}$ where $g = k\pi/2$. Note that fixing a single value of $\theta$ gives Example 4.11; in other words, this experiment has infinitely many orbits where each orbit is similar to Example 4.11.

We generate random polynomial function $f$ that is not equivariant, i.e., $\exists(\theta, x)$ s.t. $g \cdot f(\theta, x) \neq \rho_Y(g)y$. Then we try to fit $f$ using a $G$-invariant network and a $G$-equivariant network. We measure their error compared with the theoretical lower bound given by Theorem 4.8 and 4.9. As is shown in Table 2, both the invariant network and the equivariant network achieve an error rate nearly the same as our theoretical bound. The empirical error is slightly higher than the theoretical error due to the neural network fitting error. Please refer to I.4 for more experiment details.

# I    Experiment Details

This section describes the details of our experiments. All of the experiment is performed using a single Nvidia RTX 2080 Ti graphic card.

## I.1    Swiss Roll Experiment

In the Swiss Roll Experiment in Section 6.1, we use a three-layer MLP for the unconstrained network. For the $z$-invariant network, we use a network with two DSS [37] layers to implement the $z$-invariance, each containing two FC layers. We train the networks using the Adam [24] optimizer with a learning rate of $10^{-3}$. The batch size is 128. In each run, there are 200 training data, 200 validation data, and 200 test data randomly sampled from the data distribution. The network is trained for a minimal of 1000 epochs and a maximum of 10000 epochs, where the training is terminated after there is no improvement in the classification success rate in the validation set for a consecutive of 1000 epochs. We report the test success rate of the epoch model with the highest validation success rate.

## I.2    Square Experiment

In the Square Experiment in Section H.2, we use a network with two DSS [37] layers to implement the horizontal invariance, where each layer contains two FC layers. We train the networks using the Adam [24] optimizer with a learning rate of $10^{-3}$. The batch size is 128. In each run, there are 1000 training data, 200 validation data, and 200 test data randomly sampled from the data distribution. The network is trained for a minimal of 1000 epochs and a maximum of 10000 epochs, where the training is terminated after there is no improvement in the classification success rate in the validation set for a consecutive of 1000 epochs. We report the test success rate of the epoch model with the highest validation success rate.

## I.3    Digit Classification Experiment

In the Digit Classification Experiment in Section 6.2, we use two similar five-layer convolutional networks for the $D_4$-invariant network and the CNN, where the $D_4$-invariant network is implemented using the e2cnn package [60]. Both networks have the similar amount of trainable parameters. We train the networks using the Adam [24] optimizer with a learning rate of $5 \times 10^{-5}$ and weight decay

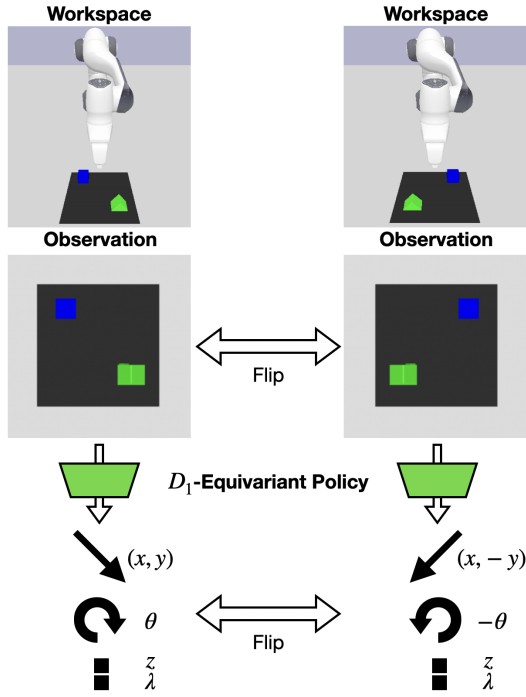

Figure 15: The robotic experiment setup and the $D_1$-equivariant policy network.

of $10^{-5}$. The batch size is 256. In each run, there are 5000 training data, 1000 validation data, and 1000 test data randomly sampled from the data distribution. The network is trained for a minimal of 50 epochs and a maximum of 1000 epochs, where the training is terminated after there is no improvement in the classification success rate in the validation set for a consecutive of 50 epochs. We report the test success rate of the epoch model with the highest validation success rate.

## I.4 Regression Experiment

In the regression experiment, we validate our theoretical error lower bound for invariant and equivariant regression (Theorem 4.8, 4.9) by comparing empirical network fitting error and the theoretical fitting error of a function $f$. Specifically, the function $f$ maps a distance $\theta$ and an index $x$ pair to a vector $y$:

$$f : \mathbb{R} \times \mathcal{X} \to \mathbb{R}^2, \text{given by } (\theta, x) \mapsto y \tag{25}$$

where $\mathcal{X} = \{x_0, x_1, x_2, x_3\}$. The group $g \in G = C_4 = \{e, r, r^2, r^3\}$ acts on $(\theta, x)$ by $g(\theta, x) = (\theta, gx)$ through permuting the index $x$: $x_1 = rx_0; x_2 = rx_1; x_3 = rx_2; x_0 = rx_3$. Let $r^k \in G$ acts on vector $y$ by rotation $\rho_Y(g) = \begin{pmatrix} \cos g & -\sin g \\ \sin g & \cos g \end{pmatrix}$ where $g = k\pi/2$.

We construct function $f$ in the following way: for each $x \in \mathcal{X}$, choose $l_x : \mathbb{R} \to \mathbb{R}^2$ and define $f(\theta, x) = l_x(\theta)$. Notice that when $l_{gx} = \rho_Y(g)l_x(\theta)$, $f$ is $G$-equivariant. We define $l_x(\theta) = (p_x(\theta), q_x(\theta))$ where $p_x$ and $q_x$ are cubic polynomials of $x$, i.g., $p_x$ with coefficients $a, b, c, d$ will be $p_x = ax^3 + bx^2 + cx + d$. We choose $p_x$ and $q_x$ with different coefficients for each $x$ such that $f$ is not equivariant, i.e., $l_{gx} \neq \rho_Y(g)l_x(\theta)$. For each run, we generate a function $f$, sample data $\theta, x$, and evaluate the data obtaining $y$. Then we train neural networks using $L2$ loss till converge. Eventually, we sample another set of data to evaluate the empirical $L2$ error as well as the theoretical $L2$ error.

## I.5 Robotic Experiment

In the robotic manipulation experiment, the state $s$ is defined as a top-down RGBD image of the workspace centered at the gripper's position (Figure 15 middle). The action $a = (x, y, z, \theta, \lambda)$ is defined as the change of position $(x, y, z)$ and top-down orientation $(\theta)$ of the gripper, with the

gripper open width ($\lambda$). For a $D_1 = \{1, g\}$ group where $g$ represents a horizontal flip, the group action on the state space $gs$ is defined as flipping the image; the group action on the action space $ga$ is defined as flipping the $y$ and $\theta$ action and leaving the other action components unchanged, $ga = (x, -y, z, -\theta, \lambda)$. We define a $D_1$-equivariant policy network $\pi : s \mapsto a$ using e2cnn [60], where the output action of $\pi$ will flip accordingly when the input image is flipped (Figure 15 bottom). We train the network using the Adam [24] optimizer with a learning rate of $10^{-3}$ and weight decay of $10^{-5}$. For each run, we train the network for a total of 20k training steps, where we perform evaluation for 100 episodes every 2k training steps. We report the highest success rate of the 10 evaluations as the result of the run.

We develop the experimental environments in the PyBullet [12] simulator, based on the BullatArm benchmark [57]. In the Stacking (correct equivariance) and Sorting (incorrect equivariance) experiment, we gather a total of $400$ episodes of demonstrations, where $400c$ of them are Stacking and the rest $400(1 - c)$ are Sorting. In evaluation, the task follows the same distribution, where $100c$ of the evaluation episodes are Stacking and the rest are Sorting. Notice that the agent can distinguish the Stacking and Sorting tasks because the object colors are different for the two tasks (green and blue for stacking, yellow and red for sorting). In the Sorting (extrinsic equivariance) experiment, we also use 400 episodes of demonstrations.

Specifically, in Sorting, the cube and the triangle are initially placed randomly, within a distance of $\pm 1.5 cm$ from the horizontal mid-line of the workspace. The objective is to push the triangle at least $9cm$ toward left and to push the cube at least $9cm$ toward right, while ensuring that both objects remain within the boundaries of workspace. In Stacking, two blocks are randomly initialized on the floor of the workspace. The goal is to pick up the triangle and place it on top of the cube. The workspace has a size of $30cm \times 30cm \times 25cm$.

