# OpenReview forum: "A General Theory of Correct, Incorrect, and Extrinsic Equivariance"
_NeurIPS.cc/2023/Conference — NeurIPS 2023 poster_

### Official Review · Reviewer_c3pV · 2023-07-07

**Soundness:** 2 fair
**Presentation:** 2 fair
**Contribution:** 2 fair
**Rating:** 5
**Confidence:** 3

**Summary:**

The authors analyze how equivariant models behave under mismatches between the symmetries of the model and data distribution.  They advance three notions, correct, extrinsic, and incorrect pointwise equivariance.  They then propose bounds on the error for equivariant neural networks that are sensitive to the data distribution and give an example where having an equivariant model can hurt learning performance.

(Edit, 8/13: in light of comments by the Authors, I have changed my score to a 5)

**Strengths:**

Theoretically, the bounds present are more general than those proposed in Wang et al. 52.  The examples clearly demonstrate the underlying mathematical principles involved.  Moreover the error bounds presented, while not unsurprising to practitioners more versed in group theory, seem mathematically solid.  On the whole, the task of attempting to precisely characterize how the data distribution interacts with symmetries imposed on the model is a worthy one.

**Weaknesses:**

While the paper is solid, I think certain difficulties in presentation prevent it from fulfilling its potential.  Most immediately, I think there are some issues in the presentation in the concepts of pointwise correct / incorrect / extrinsic equivariance.  First, equivariance is a property of maps rather than spaces: saying a point has correct / incorrect / extrinsic equivariance, or even equivariance of any kind, isn’t precise.  It would be better to say something the lines of “h has correct / incorrect / extrinsic equivariance with respect to f and p at x”, particularly since the notion of correctness here is dependent on the model: this has the advantage of being a more close parallel with Defs 3.1-3.3. Second (forgive me if I missed it), but at no point is the notion of pointwise equivariance being dependent on specific group elements used in the rest of the paper.  It is not clear what is gained by this added generality.

Continuing on to Section 5, it is not clear to me to what extent the notions advanced in section 4 are necessary for stating these bounds: if I understand correctly the bounds presented are for general probability distributions and are independent of the definitions proposed in 4.1-4.3.

Section 6 seems a bit trivial for me: it seems obvious to me that if you can’t represent a decision boundary due to a constraint in the model you might end up having a bad day.  Indeed, the specific example shown here seems to follow straightforwardly from https://doi.org/10.48550/arXiv.2110.07472.  I think this section should be moved up and be a motivating example for the bounds in Section 5.

I feel like the meat of the paper is the error bounds, more specifically Theorem 5.8.  I’m not convinced the concepts in Definitions 4.1-4.3 contribute much towards the construction of this bound.  I would recommend a rewrite of the paper that focuses more on the bounds, their implications and applications, and maybe expand further.

Finally, (and this might just be me being a pedant), but I’m not sure the title really is reflective of the content of the paper.  It is not clear to me what the general theory is that is being advanced.  Typically a “general theory” is a precise characterization of a class of mathematical objects.  If this is intended to refer to the decomposition into correct / incorrect / extrinsic equivariance, then this characterization seems to reduce to the fact that that equalities can be true, not true, or involve the empty set.  I don’t think any insight is gained innately from this characterization, rather than this being a useful language to describe further results in the text.  The error bounds, while good contributions, are specific to certain choices of loss function.  I mention this not to diminish their contributions, merely to observe that they are not “general” in a mathematical sense.


(edited for some spelling mistakes, although I'm sure more of them abound...)

**Questions:**

I’m having a little trouble parsing the notion of Fundamental domain in Definition 5.1.  What happens if the orbit of the passes over X multiple times.  For instance, consider the real line, where the group is S_8 and the group action is the antisymmetric representation (reflect the line on odd permutations)?

**Limitations:**

The limitations of the work have been appropriately addressed.

---

> ### Author Rebuttal · Authors · 2023-08-09
>
> The Authors thank the reviewer for their insightful review. Please see our response below.
>
> > equivariance is a property of maps rather than spaces...
>
> This is a good point, we definitely agree with the reviewer that ‘equivariance’ is a property of maps rather than spaces. In the revision, we will revise the definitions based on the reviewer’s suggestions to make them more precise.
>
> > at no point is the notion of pointwise equivariance used here being dependent on specific group elements used in the rest of the paper...
>
> > it is not clear to me to what extent the notions advanced in section 4 are necessary for stating these bounds...
>
> The pointwise equivariance is actually used in Proposition 5.4 as a way of calculating the error bound for the classification task. Later on, we also used it in Example 5.5 and the experiment in Section 7.1. The pointwise definitions are necessary for understanding our lower bounds because, intuitively, the bounds calculate the error of the model where pointwise incorrect equivariance exists.
>
> > Section 6 seems a bit trivial...
>
> We agree that the discussion of Section 6 is not overly complicated. However, we believe it is still important to complete our theory, as extrinsic equivariance can be another source of failure for an equivariant model beyond the incorrect equivariance discussed in Section 5. Moreover, while incorrect equivariance is often considered in the prior works, the effect of extrinsic equivariance is under explored. The prior work by Wang et al. [52] showed that extrinsic equivariance is helpful in all of their experiments. Just by reading [52], one might think that extrinsic equivariance is always helpful, but it is not true. Section 6 provides a discussion about the scenarios when extrinsic equivariance can be harmful.
>
> > Indeed, the specific example shown here seems to follow straightforwardly from https://doi.org/10.48550/arXiv.2110.07472.
>
> Although the example In Figure 1 of https://doi.org/10.48550/arXiv.2110.07472 [A] is similar to our example in Section 6, it does not directly follow. Figure 1 in [A] demonstrates that there exists a subspace that is fixed by the group action and that it affects the perceptron capacity. While they consider separating invariant orbits of data samples, our example shows an extrinsically equivariant scenario where the C2 group action produces samples outside of the data domain. For example, the group action transforms $x=(0,0,-1)$ to $gx=(0, 0, 1)$, which is outside of the set of 4 points in Figure 5a and so the symmetry relates in-distribution samples to points that are out-of-distribution. We will clarify these differences in the main text.
>
> >I think this section should be moved up and be a motivating example for the bounds in Section 5.
>
> Unfortunately, we respectfully disagree that Section 6 can be moved up to motivate Section 5, as they discuss the error caused by two different types of equivariance: Section 5 discusses incorrect equivariance and Section 6 discusses harmful extrinsic equivariance. Although both of them can cause the model to have an unsatisfactory performance, the underlying reasons are different. Specifically, incorrect equivariance leads to an inevitable error in the model due to the equivariant constraint, and we can calculate the theoretical lower bound of such an error. On the other hand, harmful extrinsic equivariance increases the complexity of the task, but unlike incorrect equivariance, it can be solved by increasing model capacity.
>
> >I feel like the meat of the paper are the error bounds...
>
> This is a good suggestion, thanks for pointing this out. We agree that the most important contribution of the paper is Section 5, rather than Section 4. In the revision, we propose to make the following changes to our paper:
> 1. We will rework the story to focus more on the contribution of the paper: the cases where equivariant models might underperform due to approximation errors and harmful extrinsic equivariance.
> 2. We plan to merge Section 3 and Section 4 into a combined background and preliminaries section.
>
> >I’m not sure the title really is reflective of the content of the paper...
>
> Thank you for the comment. Our paper builds heavily upon [52] where the authors defined the terms of correct, incorrect, and extrinsic equivariance. However, their discussion is not complete as they assume that the data density over the domain is group invariant and that the group is a finite group. Our work generalizes theirs by removing those simplifying assumptions, discussing the lower bounds for regression (rather than just classification as in the prior work), and discussing the possibility of extrinsic equivariance being harmful. Thus, we named our paper “A General Theory”. Nevertheless, we don’t mean to misuse the term “general” in a mathematical sense, and we are happy to remove the word “general” in the title of our paper.
>
> >I’m having a little trouble parsing the notion of Fundamental domain...
>
> Generally, if the orbit passes over $X$ multiple times, the stabilizer will be non-empty, and the factor $\alpha(x, g)$ in Equation 3 will account for the over-counted conjugates. In the example, the group $S_8$ acts on $\mathbb{R}$ s.t. $s_i \cdot x = -1^{i}x$ for $x\in \mathbb{R}, s_i\in S_8$, the fundamental domain $F$ would be either $\\{x>=0 | x\in\mathbb{R}\\}$ or $\\{x<=0 | x\in\mathbb{R}\\}$. Notice that this does not violate our assumption in line 161 because $\cup_{g_1F \neq g_2F} (g_1F \cap g_2F) = \\{0\\}$ has 0 measure under $\nu$. Since $S_8$ is a finite group, Equation 3 becomes $k(Gx) =\min_{y\in Y} \sum_{g\in S_8}p(gx) \mathbb{1} (f(gx) \neq y) \alpha(x, g) dg$, where $\alpha(x, g)=\frac{2}{8}$ will account for the over-counted conjugates.

---

> > ### Comment · Reviewer_c3pV · 2023-08-13
> > **Response to the Response**
> >
> > Thank you to the authors for their detailed response to my comment.  I still have reservations, but in light of proposed changes and the feedback of other reviewers, I am happy to raise my score to a 5.
> >
> > A lot of the nature of the feedback is related to the question of, "how straightforward or how non-obvious are certain elements."  I'll discuss this more in my response the other reviewers above.
> >
> > There are two technical points I wanted to follow up on.
> >
> > > The pointwise equivariance is actually used in Proposition 5.4 as a way of calculating the error bound for the classification task. Later on, we also used it in Example 5.5 and the experiment in Section 7.1. The pointwise definitions are necessary for understanding our lower bounds because, intuitively, the bounds calculate the error of the model where pointwise incorrect equivariance exists.
> >
> > There are two issues at play here.  The first is, "is p-wise equivariance necessary *in section 5*"  It definitely is not in proposition 5.4 (which holds far an arbitrary density), and I maintain that ex. 5.5 could easily be rewritten to not use these definitions.  This is not a criticism of the contribution of the bounds though: rather, it is me saying that the bounds in Section 5 are actually general for *all* probability densities, not just for incorrect equivariance as stated in the paper.  My proposed (but only suggested) change is that the authors be bolder at selling the bounds, as they are a separate co-equal contribution to the theoretical concepts in section 3: currently, I think and importance of the bounds is undersold.
> >
> > The second is, "at any point in the paper, is the fact that of p-wise equivariance depends *on a specific group element $g$* used?  As far I can tell it is not: only the dependence on $x$ is used, even in the examples mentioned by the authors.  Consequently, it should be possible to make more specific definitions that only have dependence on $x$.  However, I recognize this might be more of a style thing, and changing the definition at this point may require rewrites).
> >
> > > Generally, if the orbit passes....
> >
> > Ok, this is how I thought the fundamental domain worked.  But in this case, isn't the assumption that intersection of any two conjugates have measure 0 violated?  As far as I can tell, the authors use the word "conjugate" to mean a set as $g F$ or some $g$ in the group.  (I might be wrong about this: the use of conjugate in this paper is not the typical one in group theory, and I don't see a definition in the paper).  In the example we are discussing, the conjugate of the group elements $e$ and of any even permutation have as their conjugate {$x \geq 0 | x \in R$}.  So, by {assumption $x \geq 0 | x \in R$} has measure 0.  By similar arguments, the same is true for {$x < 0 | x \in R$}.  But this suggests that, by countable additivity, the entire real line has zero measure, which seems like a problem to me.
> >
> > Did I misunderstand something?  If not, their use of fundamental domain could noticeably restrict the applicability of their results, but I think this could be fixed without too much effort by modifying the non-intersection assumption to only hold for coset representatives for cosets of the stabilizer subgroup.
> >
> > *edit: I just saw the Author's response above giving their definition of conjugate, which makes me more confident in the analysis above.

---

> > > ### Author Response · Authors · 2023-08-15
> > >
> > > The authors appreciate the reviewer’s follow up discussion and the score increase. Please see our response below.
> > > >The first is, "is p-wise equivariance necessary in section 5" It definitely is not in proposition 5.4 (which holds far an arbitrary density), and I maintain that ex. 5.5 could easily be rewritten to not use these definitions. This is not a criticism of the contribution of the bounds though: rather, it is me saying that the bounds in Section 5 are actually general for all probability densities, not just for incorrect equivariance as stated in the paper. My proposed (but only suggested) change is that the authors be bolder at selling the bounds, as they are a separate co-equal contribution to the theoretical concepts in section 3: currently, I think and importance of the bounds is undersold.
> > >
> > > First, we would like to kindly point out that pointwise equivariance is indeed used in Proposition 5.4: the integral is $\int_G p(gx') \mathbb{1}((x',g)\in I)\alpha(x', g) dg$, where $I$ is the set of pointwise incorrect equivariance. However, you are right that both Proposition 5.4 and Example 5.5 can be rewritten to not use the pointwise definitions (for Proposition 5.4, the version that does not use the pointwise definitions is Theorem 5.3.). We also agree that the bounds in Section 5 are general for all probability densities. Notice that we set 5.3 as a Theorem but 5.4 as a Proposition exactly because 5.3 is a more general form and does not depend on the pointwise definitions.
> > >
> > > The pointwise equivariance is the source of the error bounds: if there is no pointwise incorrect equivariance, the error bounds would just be 0. However, we agree that Section 5 (without 5.3 and 5.4) can be mathematically correct without the pointwise definitions at all. In other words, Section 4 and Section 5 are connected, but not dependent on each other. We realize that the current paper might look like Section 5 entirely depends on Section 4, and the importance of the bounds is undersold. We appreciate the reviewer's suggestion for being bolder at selling the bounds, and will revise the paper to strengthen the importance of the bounds.
> > >
> > > > The second is, "at any point in the paper, is the fact that of p-wise equivariance depends on a specific group element $g$ used? As far I can tell it is not: only the dependence on $x$ is used, even in the examples mentioned by the authors. Consequently, it should be possible to make more specific definitions that only have dependence on $x$. However, I recognize this might be more of a style thing, and changing the definition at this point may require rewrites).
> > >
> > > The authors thank the reviewer for the question. The pointwise equivariance depending on a specific group element is used in Proposition 5.4, where we have the integral of the incorrect equivariance over all group elements. When we were developing the paper, we tried to make the pointwise equivariance depend on $x$ only, but could not find a satisfactory definition since we cannot evaluate the equivariance of the model on a particular $x$ without referencing its transformations. If we define the pointwise equivariance on $x$ only, there needs to be some averaging over the group, which will lose some information.
> > >
> > > Regarding the conjugates, the authors appreciate the reviewer’s thoughtful response. In the paper, when we say "the intersection of any two conjugates has 0 measure under $\nu$", we meant "two distinct conjugates" ($g_1F \neq g_2F$, not necessarily $g_1 \neq g_2$). This is spelled out on Line 161, but not in the definition. We will be sure to clarify that in the paper.  This agrees with your suggestion to modify the “non-intersection assumption to only hold for coset representatives for cosets of the stabilizer subgroup.”
> > >
> > > The authors thank the reviewer again for the insightful discussion, please let us know if our response addresses your concerns.

---

### Official Review · Reviewer_yJf8 · 2023-07-07

**Soundness:** 3 good
**Presentation:** 3 good
**Contribution:** 3 good
**Rating:** 6
**Confidence:** 2

**Summary:**

This paper provides a general theory of correct, incorrect, and extrinsic equivariance of functions, mainly extending the framework of Wang et al., 2023 to more general case of pointwise equivariance of functions defined for pairs of group element and input data. The theory mainly concerns deriving lower bound of errors for classification and (invariant and equivariant) regression tasks. The authors further identify cases where extrinsic equivariance can be harmful for performance opposed to the empirical observations of Wang et al., 2023. The authors provide a range of experiments mainly on empirically verifying the derived lower bound of errors, and also demonstrating the cases where certain extrinsic equivariances can be harmful.

Wang et al., The surprising effectiveness of equivariant models in domains with latent symmetry (2023)

**Strengths:**

S1. The theory proposed in the paper is indeed quite general as it considers pointwise equivariance, and covers a large range of partial symmetries. The fact that it addresses a wide range of tasks (classification, invariant regression, equivariant regression) also strengthens the generality of the paper.

S2. The paper is overall well written, with intuitive illustrations on pointwise equivariance and experimental setups, as well as results.

S3. The experimental results support the main claims of the paper on error bounds and harmful cases of extrinsic equivariances.

**Weaknesses:**

W1. A weakness of this work is that, while certain cases are presented where extrinsic equivariances can be harmful, it offers little principled understanding or general theory of in which specific cases extrinsic equivariance is harmful or beneficial.

W2. While the presented theory on pointwise equivariance covers a wide range of approximate or misspecified symmetries, I am not sure if it is immediately useful, since in many applications involving approximately or misspecified equivariant neural networks, we would not be able to exactly know the extent of correct, incorrect, and extrinsic pointwise equivariance of the model (unlike the synthetic experimental setups considered in the paper). This could be a weakness of the proposed theoretical framework.

**Questions:**

I have no particular questions, but would like to hear the opinions of the authors on the aforementioned weaknesses.

**Limitations:**

The authors have identified the limitation of the work in Section 8.

---

> ### Author Rebuttal · Authors · 2023-08-09
>
> The authors thank the reviewer for their helpful comments. Please see our response below.
>
> > W1. A weakness of this work is that, while certain cases are presented where extrinsic equivariances can be harmful, it offers little principled understanding or general theory of in which specific cases extrinsic equivariance is harmful or beneficial.
>
> You are right, in the context of our paper, we only show the possibility of extrinsic equivariance being harmful, instead of providing an understanding of when it would be harmful or helpful (in part, this is because the prior work [52] shows helpful extrinsic equivariance and we are trying to do a counterpart). This is a limitation of our work and a very important future direction. We will address this in the limitation section.
>
> > W2. While the presented theory on pointwise equivariance covers a wide range of approximate or misspecified symmetries, I am not sure if it is immediately useful, since in many applications involving approximately or misspecified equivariant neural networks, we would not be able to exactly know the extent of correct, incorrect, and extrinsic pointwise equivariance of the model (unlike the synthetic experimental setups considered in the paper). This could be a weakness of the proposed theoretical framework.
>
> We agree that in many real-world applications, we might not exactly know the extent of correct, incorrect, and extrinsic pointwise equivariance of the model and it might be hard to compute the lower bound. However, our analysis also provides a theory that can guide the model selection process in such a case. Equivariant models are usually selected on the basis of prior reasoning about properties of the ground truth function. We believe our theory provides intuition which can be part of this prior reasoning process and can be used to understand and analyze model performance in an iterative model development cycle. Moreover, in some real-world applications, we can indeed analyze the extent of different equivariance types. For example, in robotic manipulation, we might know a priori how the objects are distributed on a tabletop, so we can analyze how rotations of the entire workspace change the distribution.

---

> > ### Comment · Reviewer_yJf8 · 2023-08-18
> > **Response to rebuttal**
> >
> > Thank you for the response. I recommend the authors to supplement the discussions on limitations in Section 8 with the provided response. I have no further questions for now.

---

### Official Review · Reviewer_bWzk · 2023-07-07

**Soundness:** 3 good
**Presentation:** 3 good
**Contribution:** 3 good
**Rating:** 7
**Confidence:** 4

**Summary:**

An obvious limitation of equivariant networks is the assumption that the symmetry they hard-code matches the symmetry of the underlying ground truth function exactly. What happens when the symmetry is only partially present in the domain affecting a symmetry mismatch between the ground-truth function and the neural network? This paper presents some systematic analysis of this situation building on prior work by Wang et al. 2023.

Wang et al. classified models as having "correct" equivariance (the underlying function provided by nature matches the symmetry of the equivariant model), "incorrect" equivariance (there is a disagreement between the underlying true function and that encoded by the equivariant network), and "extrinsic" equivariance (model symmetry transforms in-distribution data to out-of-distribution data). This paper defines these notions pointwise which allows one to generalize to a continuum of equivariance types when some tasks may have different proportions of these three situations (and does not exclusively conform to only one of them). Lower bounds are provided on the model error from incorrect model symmetry (while improving some earlier results) for classification. Morally similar lower bounds are provided for the regression case in terms of the variance of the function to be modelled over the orbit of the group under consideration. It is also shown that extrinsic equivariance can be harmful (which doesn't contradict but can still be contrasted with earlier experimental results in the literature).

The theoretical results are provided in section 5 (after first introducing the notions of correct, incorrect and equivariance of Wang et al, and then stating their pointwise generalizations which open the path to interpolate between them -- a central aim of the paper). The lower bound for classification is intuitive and basically says that it is equal to the integral of the total dissent over the fundamental domain. The total dissent measures how many elements in the orbit of G have a different label than the majority label. Two examples are given for binary and multiclass cases to build intuition (also illustrated in figures 3 and 4). Next, the case of invariant regression is treated. The result is similar in spirit to the result from earlier -- the error is bounded by an integral over the fundamental domain of p(Gx) times the variance of the function on the orbit Gx (so instead of the dissent, we look at the variance). A slightly more careful consideration permits an easy generalization for equivariant regression. A simple theoretical argument is then provided to show that extrinsic can be harmful for generalization, also raising the question that it remains open to understanding its effect on generalization.

**Strengths:**

- The paper considers an important problem that has only recently started getting treated in the literature. It is clear that equivariant networks make a strong assumption about the symmetry of the ground truth function. However, it is not clear how much a mismatch matters. This paper provides some approximation lower bounds due to the mismatch for some different settings.
-  Being able to interpolate between the three types can definitely help improve model generalization and permit more flexible models.
- While many of the points raised in the paper seem somewhat obvious at times, and computing the lower bound might not be possible in practice, the presented work can still give some guidance on model selection.

**Weaknesses:**

- Somewhat fortuitously, I was a reviewer for this paper for an earlier iteration. Whatever weaknesses that I had raised at the point seem to have been adequately addressed by the authors (in fact they were right then). I also notice that the other weaknesses bought up by other reviewers at the time (mostly in terms of experiments, and comparison to the work of Wang) have also mostly been incorporated. While I can nitpick, I have no hesitation in simply voting for acceptance.

**Questions:**

No

**Limitations:**

Yes

---

> ### Author Rebuttal · Authors · 2023-08-09
>
> The authors thank the review for their careful review. We are glad that the reviewer acknowledges that we have addressed the problems from the earlier iteration. If you have any other questions regarding our paper, please don’t hesitate to let us know and we are more than happy to discuss them.

---

### Official Review · Reviewer_racn · 2023-07-11

**Soundness:** 3 good
**Presentation:** 3 good
**Contribution:** 2 fair
**Rating:** 7
**Confidence:** 2

**Summary:**

The paper analyzes error bounds for models constrained to satisfy symmetries that only partially agree with the ground truth functions. The main suggestion is to generalize previous definitions, to the point level, allowing to derive a lower bound on the model error with respect to the volume of the portion of the domain where symmetries are mismatched. The theoretical analysis is validated on some toy examples and some (relatively) small-scale datasets

**Strengths:**

The proposed analysis including the definitions, propositions, and theorems seems to be simple.

The paper is well-written and easy to follow. I appreciate the illustrations and toy examples provided.

The experiments seem to support the theoretical analysis. Where mismatches occur, reasonable explanations are provided.


**Weaknesses:**

Missing discussion

I feel the text should elaborate more on the assumptions taken in the analysis of error bound. For example, it is assumed that the model assigns labels by taking a majority vote in an orbit. How reasonable is this assumption in relation to existing equivariant models?

In addition, it would be interesting to incorporate a model that approximates the majority vote by sampling for one of the experiments.  E.g., does it improve the digit classification model?


**Questions:**

No further questions.

**Limitations:**

Yes

---

> ### Author Rebuttal · Authors · 2023-08-09
>
> The authors thank the reviewer for their thoughtful review. Please see our response below.
>
> > I feel the text should elaborate more on the assumptions taken in the analysis of error bound. For example, it is assumed that the model assigns labels by taking a majority vote in an orbit. How reasonable is this assumption in relation to existing equivariant models?
>
> Note that label assignment by majority vote is *not* an assumption of Theorem 5.3, which applies to any invariant $h$. We will clarify the hypothesis in the theorem statement. The $h^*$ which uses majority voting is part of the proof for Theorem 5.3. Intuitively, we compute a lower bound on $err(h)$ for any invariant $h$ by comparing it to the error of the best case invariant hypothesis $err(h^*)$. If the model were not to follow this majority voting scheme, it would make more mistakes in an orbit, resulting in a higher error (as illustrated in the Equation at the end of page 4).
>
> > In addition, it would be interesting to incorporate a model that approximates the majority vote by sampling for one of the experiments. E.g., does it improve the digit classification model?
>
> Thank you for suggesting this experiment. We interpret the proposed approach as follows: instead of learning an invariant model for the digit classification task, we will have a sample-based model. To predict the label of $x$, this model will access $f(gx)$ for all $g\in G$ and calculate the majority vote as the output. This is a very interesting approach, but we think it might not be practical in our experiment because it will need access to an oracle ground truth function $f(x)$ during evaluation. Please let us know if we didn’t interpret your suggestion correctly.

---

### Official Review · Reviewer_1dq7 · 2023-07-24

**Soundness:** 3 good
**Presentation:** 3 good
**Contribution:** 2 fair
**Rating:** 5
**Confidence:** 4

**Summary:**

The paper presents an extension of a lower bound on error in finite labeled classification introduced in "The Surprising Effectiveness of Equivariant Models in Domains with Latent Symmetry" and investigates lower bounds on error for G-equivariant and G-invariant regression, offering valuable insights into the performance of symmetry-preserving models in regression tasks.

**Strengths:**

This extension holds significant practical importance, particularly when considering real-world natural symmetry groups such as $G = SO(3)$, which go beyond the scope of the original paper. Moreover, the authors have done an excellent job in presenting the material in an accessible manner, making the paper easy to read and well-structured.

**Weaknesses:**

1.
The paper's focus on pointwise definitions rather than emphasizing their implications may obscure the novelty of the results, potentially hindering a clear understanding of the significance of their contributions.

2.
While the experiments conducted in the paper offer valuable insights, their omission of infinite groups may limit the demonstration of the full strength of the new lower bound. Additionally, the observations suggesting equivariant models' potential ineffectiveness in classical learning tasks raise valid questions about their relevance and applicability.

3.
Although the paper draws upon certain notions from "The Surprising Effectiveness of Equivariant Models in Domains with Latent Symmetry," it does so only partially. This partial utilization of concepts, coupled with the use of different evaluation metrics such as "accuracy" in the original paper and "error" in this work, may create challenges when attempting to directly compare the results between the two studies.

**Questions:**

1.
Can you elaborate on the third experiment? Which state is flipped? (line 532, page 9)

2.
Did you mean "orbits" instead of conjugates in the definition of the fundamental domain? (line 158, page 4)

**Limitations:**

The author did address the limitations of reflection symmetries on the expressivity of models in Chapter 6 (line 254, page 7), and the requirement of knowledge about the density function (line 386, page 9)

---

> ### Author Rebuttal · Authors · 2023-08-09
>
> The authors appreciate the reviewer’s insightful comments and questions. Please see our response below.
>
> > The paper's focus on pointwise definitions rather than emphasizing their implications may obscure the novelty of the results, potentially hindering a clear understanding of the significance of their contributions.
>
> We appreciate your feedback. The pointwise definitions are a generalization of the prior work [52]. They help us analyze the problem of incorrect/extrinsic equivariance more completely and can be used to calculate the lower bound, as is shown in Proposition 5.4. However, the major contribution of our work is the bounds in Section 5 instead of the pointwise definitions. We plan to merge Section 4 into Section 3 and iterate on our writing to make the contributions clearer.
>
> > While the experiments conducted in the paper offer valuable insights, their omission of infinite groups may limit the demonstration of the full strength of the new lower bound.
>
> Thanks for the comment. While our theoretical findings are not restricted to finite groups, we opted to use finite groups in our examples and experiments for clarity and ease of understanding.
>
> > Additionally, the observations suggesting equivariant models' potential ineffectiveness in classical learning tasks raise valid questions about their relevance and applicability.
>
> Equivariant models have been shown to be effective in many learning tasks. However, given the constrained nature of equivariant models, it is possible that they will lead to inevitable errors in some cases. Our study explores these potential scenarios of ineffectiveness, which we believe can provide valuable insights for model selection and analysis.
>
> > Although the paper draws upon certain notions from "The Surprising Effectiveness of Equivariant Models in Domains with Latent Symmetry," it does so only partially. This partial utilization of concepts, coupled with the use of different evaluation metrics such as "accuracy" in the original paper and "error" in this work, may create challenges when attempting to directly compare the results between the two studies.
>
> Our work builds upon the prior work “The Surprising Effectiveness of Equivariant Models in Domains with Latent Symmetry” to extend a theory that the prior work only partially addresses. Consequently, some of their definitions and framework are too limited and make too many assumptions. Removing these assumptions and generalizing that work is a key part of our  goal. For example, the prior work limits their analysis in classification tasks, so using “accuracy” makes sense in their context. However, our work aims to calculate a more accurate bound for both classification and regression, thus “error” would be a better choice.
>
> > Can you elaborate on the third experiment? Which state is flipped? (line 532, page 9)
>
> Thanks for asking this question. The state is a top-down RGBD image of the workspace, and by flipping the state, we mean that the image is flipped horizontally. Similarly, flipping the action means that the (x, y) component of the robot action is flipped horizontally. Due to the page limit, those details were put in Appendix K.5 instead of the main paper.
>
> > Did you mean "orbits" instead of conjugates in the definition of the fundamental domain? (line 158, page 4)
>
> We appreciate the question. However, we indeed meant "conjugates" not "orbits". An orbit of $x\in X$ is defined as the set $\\{ gx| g\in G\\}$, whereas a conjugate $gF$ is defined as $gF=\\{gx | x\in F\\}$ (please see https://mathworld.wolfram.com/FundamentalDomain.html). We understand that this could be a point of confusion, and we will ensure to clarify the definition of a conjugate in the revision.

---

### Official Review · Reviewer_FMJP · 2023-07-25

**Soundness:** 3 good
**Presentation:** 3 good
**Contribution:** 3 good
**Rating:** 8
**Confidence:** 4

**Summary:**

This work addresses the situation in which data and model equivariance does not match exactly. It extends previous work from Wang et al., by proposing a pointwise version of their definition of correct, incorrect and extrinsic equivariance. The usefulness of these new concepts are demonstrated in three new performance lower-bounds derived from classification and regression problems. A series of examples and experiments are presented to illustrate the new proposed notions, confirm the lower-bounds derived and show that they usually seem tight in practice.

**Strengths:**

### Originality
Although it builds heavily on previous work from Wang [52], the paper presents novel theoretical concepts and results, as well as new experiments supporting them.

### Clarity
Overall, the paper is well written and structured.

### Quality
The paper presents new theoretical results with their corresponding proofs, which look sound to be (disclaimer: I have some familiarity with group theory, but I am not an expert). It also presents interesting results in a large range of experiments. The latter are all rather small/toy, but are still quite convincing IMO.


**Weaknesses:**

### Clarity

Minor:

- I think there is a mistake in the xlabel of fig 7b. Shouldn't it be "incorrect - correct"? Same for figure 7a: should be "correct - extrinsic" I guess, since when x=1 you have the highest INV model performance, which should correspond to c=1 and e=0 according to the text.
- L.363: I think you wanted to reference fig.9, not 8.

### Clarity and Quality

1. I think the definition of $p$ could be clarified. It is introduced in line 92 as the “probability density function of the domain”, so, at first reading, I thought p(x) was the “true” underlying population distribution from which both training and test examples are samples. This seems to be confirmed by line 387 in the conclusion: “our theoretical lower bounds require domain knowledge like the density function over the domain”.  But in section 6, it seems that you define extrinsic equivariance with respect to the actual examples in the training set, which is not the same thing: “F_E corresponds to an extrinsically equivariant class for **this data**” l.272

2. I also have other few questions regarding the example presented in section 6. First of all, does the data from figure 5 represent training or test data? Is the data $S$ the whole support of $p(x)$? What is the true labeling function $f$ in this example?
These elements seem important to conclude. The reason I ask is that if $f$ is indeed the “exclusive or” on (x,y) coordinates (i.e. it is indeed C2-invariant) and if the examples from figure 5a are just the training data but the test data can go outside (e.g. their symmetric elements in fig 5b could be the test set), then despite the 0% error rate of the unconstrained linear model on the training data, it would learn an incorrect labeling function and its test performance would be 50%, while the invariant model would still have a 25% error rate, which would be better.

3. If $p$ really denotes the population distribution, I wonder whether we can talk about $p$ independently of $f$ and vice-versa. From your figures 1 and 2, it seems that $f$ can be defined outside of the support of $p(x)$, and I wonder whether this makes sense. For example, if $f$ is the labeling function in a digit classification problem, what should be its output for an image of the digit “9” rotated by 90 degrees? Should it be 9 or 6?

### Originality:

Minor:

4. In the related work, some references to class-specific and instance-specific automatic data augmentation works are missing ([1, 2, 3] for example), while they are strongly related to the idea of pointwise invariance proposed in the paper.

[1] https://arxiv.org/abs/2106.13695

[2] https://arxiv.org/abs/1510.02795

[3] https://arxiv.org/abs/2206.00051

**Questions:**

See 3 questions in the weaknesses section above.

**Limitations:**

Yes, the authors mention one limitation of their work, which is the need to know $p$, which is never the case in practice.

---

> ### Author Rebuttal · Authors · 2023-08-09
>
> The authors thank the reviewer for their careful review. Please see our response below.
>
> > I think there is a mistake in the xlabel of fig 7b.
>
> We apologize for any confusion regarding the `-` in `incorrect - correct` and the other x label in Fig 7. The `-` here is not meant to denote subtraction, rather, it indicates that the distribution transitions linearly from incorrect to correct as we move along the x-axis from left to right. We appreciate your comment and will revise the label to make it more explicit by changing the x label of Fig 7a to ‘correct ratio’, and labeling the x=0 as ‘c=0, e=1’ and labeling x=1 as ‘c=1, e=0’.
>
> > L.363: I think you wanted to reference fig.9, not 8.
>
> We appreciate your careful reading. Indeed, there is a referencing error. We will correct it in the revised version.
>
> > I think the definition of $p$ could be clarified...
>
> Thank you for pointing this out. $p$ is indeed the probability density function of the domain and this description was omitted in Section 6. We consider the probability density $p$ to be uniform for this domain, where the data domain consists only of the 4 samples and $p$ is zero everywhere else. The $C_2$ group action acts along the z-axis and the transformed data contains samples that were not part of the original domain (e.g., $x=(0,0,-1), gx=(0, 0, 1)$). We will add these clarifications to the final revision.
>
> > I also have other few questions regarding the example presented in section 6. First of all, does the data from figure 5 represent training or test data? Is the data $S$ the whole support of $p(x)$? What is the true labeling function $f$ in this example? These elements seem important to conclude. The reason I ask is that if $f$ is indeed the “exclusive or” on (x,y) coordinates (i.e. it is indeed C2-invariant) and if the examples from figure 5a are just the training data but the test data can go outside (e.g. their symmetric elements in fig 5b could be the test set), then despite the 0% error rate of the unconstrained linear model on the training data, it would learn an incorrect labeling function and its test performance would be 50%, while the invariant model would still have a 25% error rate, which would be better.
>
> In this example, we do not differentiate between training and test data and focus on only the learnability of a non-equivariant/invariant function class for a specific distribution and not on its generalization. The domain is discrete where Figure 5a shows the only four possible values of x with probabilities p(x) = 1/4. As we consider the model’s capacity to fit arbitrary labelings (since we use Rademacher complexity), there is no ground truth labeling function and Figure 5a shows one instance of an assignment of labels. We consider the training set S to be all 4 data points. Section 6 serves to show that for a linear class of models, an extrinsically-equivariant/invariant model has a non-zero error rate for this data. The example you compute is correct if the test data contained the 8 points shown in Figure 5b, but we do not consider this scenario as it is not an example of extrinsic equivariance. The test and training data is only the 4 points shown in Figure 5a. We will add these clarifications to the text.
>
> > If $p$ really denotes the population distribution, I wonder whether we can talk about $p$ independently of $f$ and vice-versa. From your figures 1 and 2, it seems that $f$ can be defined outside of the support of $p(x)$, and I wonder whether this makes sense. For example, if $f$ is the labeling function in a digit classification problem, what should be its output for an image of the digit “9” rotated by 90 degrees? Should it be 9 or 6?
>
> Thank you for raising this question. In our analysis, we only consider the behavior of $f$ within the support of $p$ because the output of $f$ outside the support of $p$ will not change our bounds. Note that in Equations 2 and 3, the equations are weighted by $p$, so for x outside the support of $p$, the value of $f$ does not matter. In Figures 1 and 2, we draw $f$ outside of the support of $p$ to make it continuous for ease of understanding. If they are confusing, we are happy to iterate on them to remove the part where $p(x)=0$.
>
> In practice, there can be circumstances where defining $f$ outside the support of $p$ is meaningful. For example, $f$ can be expanded outside the support of $p$ in scenarios like a random crop data augmentation, where the output of $f$ will not change after a random crop, even though the cropped image is out of distribution. In the reviewer’s example regarding digits 6 and 9, if the rotated 6 is outside the support of $p$ (like in MNIST where the rotated 6 and 9 are different due to handwriting), $f$’s output would be undefined.
>
> > In the related work, some references to class-specific and instance-specific automatic data augmentation works are missing ([1, 2, 3] for example), while they are strongly related to the idea of pointwise invariance proposed in the paper
> [1] https://arxiv.org/abs/2106.13695
> [2] https://arxiv.org/abs/1510.02795
> [3] https://arxiv.org/abs/2206.00051
>
> Thank you for pointing out those references! Yes, those related works are strongly related to our work. First, the data augmentation methods can be viewed as learning pointwise extrinsic equivariance with respect to transformations defined by the augmentation function. Second, the instance and class-specific augmentations can also be viewed as applying data augmentation where pointwise correct or extrinsic invariance exist, while avoiding incorrect invariance. Moreover, while our current work is focused on constrained model classes, our theory can also potentially be applied to analyze these data augmentation methods. We will definitely add them to the related work section.

---

> > ### Comment · Reviewer_FMJP · 2023-08-11
> > **Answer to authors**
> >
> > Dear authors,
> >
> > Thank you very much for your thoughtful rebuttal.
> >
> > I can confidently say you have addressed all my (few) concerns. The only points I would like to slightly insist on after reading your clarifications would be:
> > 1. I think indeed that it would be nice to make clear that $S$ is the whole support from which any data can be sampled in section 6 (even though the example is very simple) ;
> > 2. I also think you should say explicitly what is $p$ and that you assume there is never a distribution shift (so $p$ is always the underlying distribution when training **and** testing). I understand now why you have drawn $f$ outside of its domain, as your figure would indeed be less clear if you had restricted it to $p(x>0)$.

---

> > > ### Author Response · Authors · 2023-08-15
> > >
> > > The authors appreciate the reviewer's helpful comments. We will be sure to update the paper based on your suggestions.

---

### Decision · Program_Chairs · 2023-09-21

**Decision:**

Accept (poster)

**Comment:**

Given a group action and a distribution on the input space, a function may or may not be equivariant when restricted to the support of the distribution (assuming in addition an action on the output space). Wang et al. [52], considering mismatches between model symmetries and data distributions, also studied a third case in which the support is not invariant to the group action. The paper considers a generalization of Wang et al. [52] to pointwise equivariance, allowing situations in which the three cases considered hold in different parts of the domain.

The reviewers appreciated the generality as well as the practical relevance of the relaxed notion of equivariance considered in the paper, so as the series of experiments provided by the authors to demonstrate the validity of lower bounds so derived. After the rebuttal phase, an attempt was made to come to a consensus based on the assumption that as section 5 offers limited technical contribution, evaluation mainly hinges on possible conceptual contributions, i.e., a natural refinement of the framework introduced by [52]. Despite the discussions with the authors, those who initially gave borderline feedback largely maintained their assessment, believing that the conceptual framework, though well-presented, may either not be sufficiently provoking or that its applicability may be severely restricted by the requirement to have access to the underlying data distribution (or a proxy thereof whose quality has not been properly discussed\established). Those who gave high scores raised, by and large, similar concerns.

Eventually, following the considerable discussion the paper has generated, the reviewers coincided on: a reasonable paper to accept if there is room.